# Co-aggregation and secondary nucleation in the life cycle of human prolactin/galanin functional amyloids

**Debdeep Chatterjee[1], Reeba S Jacob[1], Soumik Ray[1], Ambuja Navalkar[1], Namrata Singh[1], Shinjinee Sengupta[1†], Laxmikant Gadhe[1], Pradeep Kadu[1], Debalina Datta[1], Ajoy Paul[1], Sakunthala Arunima[1], Surabhi Mehra[1], Chinmai Pindi[2], Santosh Kumar[3], Praful Singru[3], Sanjib Senapati[2], Samir K Maji[1]***

[1]Department of Biosciences and Bioengineering, Indian Institute of Technology Bombay, Mumbai, India; [2]Department of Biotechnology, Indian Institute of Technology Madras, Chennai, India; [3]School of Biological Sciences, National Institute of Science Education and Research, Bhubaneswar, India

**Abstract** Synergistic-aggregation and cross-seeding by two different proteins/peptides in the amyloid aggregation are well evident in various neurological disorders including Alzheimer's disease. Here, we show co-storage of human Prolactin (PRL), which is associated with lactation in mammals, and neuropeptide galanin (GAL) as functional amyloids in secretory granules (SGs) of the female rat. Using a wide variety of biophysical studies, we show that irrespective of the difference in sequence and structure, both hormones facilitate their synergic aggregation to amyloid fibrils. Although each hormone possesses homotypic seeding ability, a unidirectional cross-seeding of GAL aggregation by PRL seeds and the inability of cross seeding by mixed fibrils suggest tight regulation of functional amyloid formation by these hormones for their efficient storage in SGs. Further, the faster release of functional hormones from mixed fibrils compared to the corresponding individual amyloid, suggests a novel mechanism of heterologous amyloid formation in functional amyloids of SGs in the pituitary.

*For correspondence:
samirmaji@iitb.ac.in

Present address: †Amity Institute of Molecular Medicine and Stem Cell Research, Amity University, Noida, India

Competing interest: The authors declare that no competing interests exist.

## Editor's evaluation

This study unravels the formation of prolactin/galanin functional amyloids and their storage in secretory granules of the anterior pituitary gland. Our understanding of the regulation of hormonal release from the pituitary gland is improved with this report. It will be of interest to the field of endocrinology, neurobiology and cancer.

## Introduction

Protein/peptide misfolding, aggregation, and amyloid formation is associated with various neurological disorders such as Alzheimer's disease and Parkinson's disease (*Wszolek et al., 2001*; *Katzman and Saitoh, 1991*). However, several studies have suggested that amyloid formation is also associated with the native biological function of the host organism. The protein fibrils formed to aid in the functionality of the host organisms are called functional amyloids (*Talbot, 2003*; *Destoumieux-Garzón et al., 2003*; *Oh et al., 2007*; *Maddelein et al., 2002*; *Ritter et al., 2005*). For example, Pmel-17 amyloid fibrils template melanin polymerization inside melanosomes (*Berson et al., 2001*; *Watt et al., 2009*; *Harper et al., 2008*; *Fowler et al., 2006*; *Huff et al., 2003*); Curli fibrils of *E. coli* support the organism to adhere to a surface and also help their colonization process inside biofilms (*Barnhart and Chapman, 2006*; *Chapman et al., 2002*). In this line, another very interesting aspect is the formation

**eLife digest** The formation of plaques of proteins called 'amyloids' in the brain is one of the hallmark characteristics of both Alzheimer's and Parkinson's disease, but amyloids can form in many tissues and organs, often disrupting normal activity. A lot of the research into amyloids has focused on their role in disease, but it turns out that amyloids can also appear in healthy tissues. For example, some protein hormones form amyloids that act as storage depots, helping cells to release the hormone when it is needed.

Normally, amyloids are made mostly of a single type of protein or protein fragment associated with a particular disease like Alzheimer's. Often, this type of amyloid promotes plaque formation in other proteins, which aggravates other diseases (for example, the amyloids that form in Alzheimer's can lead to Parkinson's disease or type II diabetes getting worse).The plaques start growing from small amyloid fragments called seeds. In mixed amyloids – amyloids made of two types of proteins – seeds made of one protein can trigger the formation of amyloids of the other protein. This raises the question, is this true for hormones? The body often releases more than one hormone at a time from the same tissue; for example, the pituitary gland releases prolactin and galanin simultaneously. However, these hormones have completely different structures, so whether they can form a mixed amyloid is unclear.

To answer this question, Chatterjee et al. first determined that, within the pituitary gland of female rats, prolactin and galanin could be found together in the same cells, forming mixed amyloids. To understand out how this happens, Chatterjee et al. tried seeding new amyloids using either prolactin or galanin. This revealed that only prolactin seeds were able to trigger the formation of galanin amyloids. Chatterjee et al. also found that the mixed amyloids could release the hormones faster than amyloids made from either protein alone. Together, these results suggest that the collaboration between these two proteins may help maintain hormone balance in the body.

Problems with hormone storage and release lead to various human diseases, including prolactinoma. Understanding amyloid storage depots could reveal new ways to control hormone levels. Further research could also help to explain more about well-studied diseases linked to amyloids, like Alzheimer's.

of functional amyloid by protein/peptide hormones (such as prolactin [PRL] and galanin [GAL]) during their storage inside the secretory granules (SGs) of pituitary (*Maji et al., 2009a*; *Jacob et al., 2016*). The amyloid formation, in this case, not only enriches the protein concentration to serve as a protease-resistant protein/peptide storage depot but is also able to release the functional monomeric proteins upon dilution and pH changes (*Maji et al., 2009a*; *Jacob et al., 2016*; *Jacob et al., 2010*; *Maji and Riek, 2010*; *Anoop et al., 2014*).

Protein/peptide aggregation and amyloid formation generally follow nucleation-dependent polymerization mechanism (*Wood et al., 1999*; *Srivastava et al., 2019*; *Eden et al., 2015*), where protein/peptide slowly associates to form aggregation competent nuclei (in the lag phase of aggregation) (*Arosio et al., 2015*; *Ghosh et al., 2013*; *Mehra et al., 2018*). Once formed, the aggregation competent nuclei further recruit the monomeric counterpart for their growth into mature amyloid fibrils (elongation phase) (*Arosio et al., 2015*; *Ghosh et al., 2013*; *Mehra et al., 2018*). The progression of aggregation eventually reaches a steady-state equilibrium between the fibrils and monomeric protein (stationary phase) (*Srivastava et al., 2019*; *Alberti et al., 2010*). Recent evidences however suggest that fragmentation/elongation and secondary nucleation may contribute to a significant decrease in the lag time of aggregation, similar to an external addition of preformed nuclei in amyloid growth reaction (seeding) (*Daskalov et al., 2021*; *Törnquist et al., 2018*; *Buell, 2019*). Although homotypic aggregation and seeding is the most favored mechanism of protein aggregation and amyloid formation, synergistic aggregation (co-aggregation) by two different proteins/peptides and heterologous seeding are also suggested to be involved in many neurodegenerative disorders (*Köppen et al., 2020*; *Nisbet et al., 2015*; *Bennett et al., 2017*; *St-Amour et al., 2018*; *Spires-Jones et al., 2017*). This is one of the possible mechanisms by which one disease aggravates the other disease such as Alzheimer's disease together with either Type 2 diabetes (*Bharadwaj et al., 2017*; *Biessels et al., 2005* or Parkinson's disease *Guo et al., 2013*; *Giasson et al., 2003*).

PRL and GAL secretion is synergistic and promoted by common secretagogues (*Koshiyama et al., 1987*; *Murakami et al., 1993*; *Wynick et al., 1998*). PRL/GAL co-storage has been also reported in the anterior pituitary of female rats or estrogen-treated male rats (*Hyde et al., 1991*). Here, we investigate the synergistic aggregation and amyloid formation by these hormones for their secretory storage and release from SGs. Indeed, both of these hormones are co-stored in the female rat pituitary and possess amyloid-like characteristics. In this study, we show that both hormones not only engage in homotypic amyloid aggregation in the presence of specific glycosaminoglycans as helper molecules but also synergistically aggregate (in absence of glycosaminoglycans) to form heterotypic amyloid containing PRL and GAL as suggested by double immunoelectron microscopy. Intriguingly, cross-seeding with PRL fibril seeds resulted in fibrillation of GAL. However, GAL fibrils could not seed PRL monomers for fibrillation. This suggests a tightly controlled regulation of hormone amyloids for their homotopic and heterotypic storage. Furthermore, our in vitro release assay showed faster release of functional monomers by heterotypic, hybrid amyloid (PRL-GAL) compared to its homotypic counterparts. This supports the storage and release are highly controlled and conserved in pituitary tissue for the optimum function to be served.

## Results
### PRL and GAL are co-stored as amyloids in SGs of the anterior pituitary of female rats

Previously, it was shown that PRL and GAL co-store in the anterior pituitary of female rats or estrogen-treated male rats (*Hyde et al., 1991*; *Koshiyama et al., 1990a*; *Koshiyama et al., 1990b*). PRL is a 23 kDa protein and consists of four-helix bundles comprising residues 14–42 (helix 1), 78–104 (helix 2), 110–138 (helix 3), and 160–194 (helix 4) (*Teilum et al., 2005*; *Keeler et al., 2003*). The helix bundles are spaced with three loop regions (*Figure 1a*). On the other hand, GAL is a small unstructured neuropeptide, which is 30 residues in length (*Evans and Shine, 1991*; *Bersani et al., 1991*; *Figure 1a*). PRL showed amyloid-prone sequences predicted by TANGO (*Fernandez-Escamilla et al., 2004*); GAL, however, did not contain any amyloid-prone sequences (*Figure 1b–c*). Since previously it was established that protein/peptide hormones (including PRL and GAL) can be stored as amyloids inside the SGs (*Maji et al., 2009a*), we tested whether PRL and GAL are colocalized in the cells of the anterior pituitary in the amyloid state or not. To examine this possibility, the immunofluorescence study of female rat pituitary was performed using anti-GAL and anti-PRL antibodies (see Materials and methods). Our results showed that PRL and GAL were substantially co-localized in the pituitary tissue (*Figure 1d*) indicating their co-storage inside the SGs. Further, to test whether both PRL and GAL remain in the amyloid state, we performed immunostaining of the pituitary tissue sections with amyloid-specific antibody OC along with Thioflavin-S (ThioS) (*Bussière et al., 2004*). Double immunofluorescence of PRL and GAL along with amyloid specific OC antibody showed strong colocalization suggesting both the hormones are in the amyloid state (*Figure 1e*), which was also observed in ThioS staining (*Figure 1—figure supplement 1*). In contrast to the female rats, when a similar study of double immunofluorescence was performed in the male rat pituitary tissues, we did not observe any co-localization of PRL and GAL (*Figure 1—figure supplement 2*), which is consistent with the earlier studies (*Hyde et al., 1991*; *Steel et al., 1989*). Altogether, our data confirmed that both PRL and GAL are co-stored as amyloid aggregates in the female rat pituitary.

### In vitro amyloid aggregation kinetics and synergistic co-fibrils of PRL and GAL

The colocalization of both PRL and GAL in the amyloid state suggested that PRL and GAL might interact with each other and might co-aggregate inside the same SGs. We hypothesized that there could be four possibilities (case 1–4) when PRL and GAL are co-stored as amyloids (*Figure 2a*). The two proteins can form separate filaments and these filaments can be incorporated into heterogeneous fibrils (case 1). They can form completely separate, homogeneous fibrils either by PRL or GAL or by both hormones (case 2). It is also possible that the PRL and GAL together can form the fibril forming unit to form heterogeneous amyloid fibrils (case 3); and lastly, the PRL monomers can simply adhere to the GAL fibrils (case 4 a) or vice-versa (case 4b).

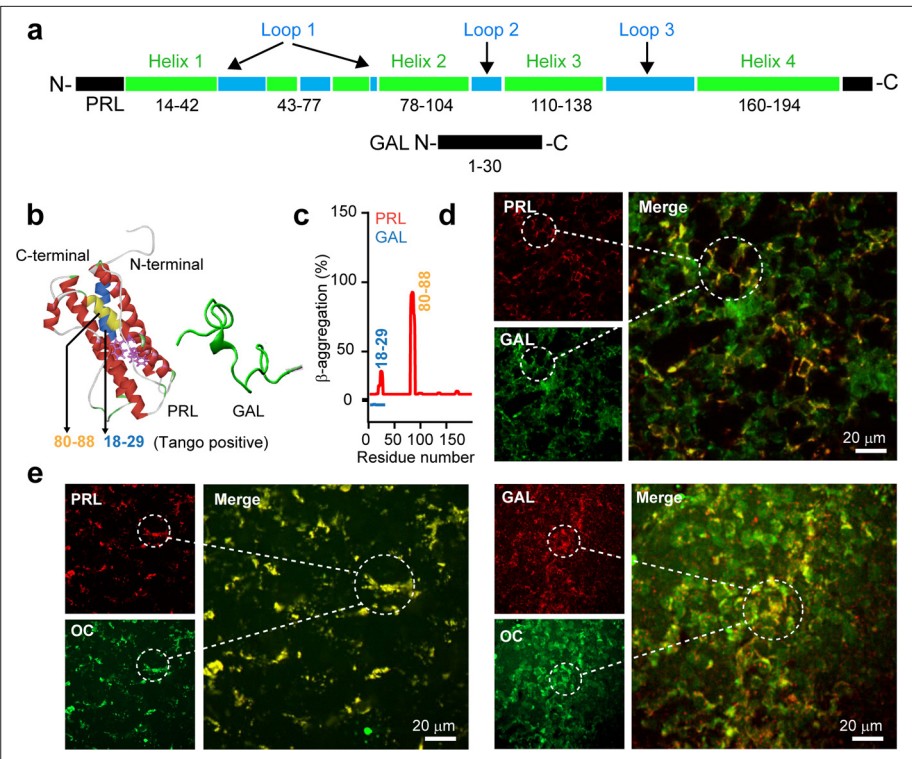

**Figure 1.** Amyloid propensity and co-storage of PRL and GAL. (**a**) Schematic showing amino acid sequence and secondary structures of PRL & GAL with different color codes. (*Upper panel*) PRL is 191 amino acids in length and contains a four-helix bundle (green). The short helix and loop regions are also represented between helix-1 and helix 2 (shown in green and blue colors, respectively). (*Lower panel*) GAL showing 30 residue peptide with no definite secondary structure (*Evans and Shine, 1991*; *Bersani et al., 1991*). (**b**) (*Left panel*) The three-dimensional structure (obtained in Pymol) (*Teilum et al., 2005*) of PRL showing its major helices and two tryptophan residues (shown in purple) (*PDB ID: 1RW5*). (*Right panel*) Natively unstructured conformation of GAL is also shown. (**c**) TANGO algorithm showing the aggregation-prone residues of PRL and GAL at pH 6.0 (SGs relevant pH). The residues 18–29 and 80–88 of PRL showing amyloid aggregation potential. However, TANGO analysis of GAL revealed no amyloid aggregation propensity. Immunofluorescence studies showing (**d**) colocalization of PRL (red) and GAL (green) in the female rat anterior pituitary. (**e**) (*left panel*) Colocalization of amyloid fibrils (OC, green) and PRL (red) and amyloid fibrils (OC, green) and GAL (red) (*Right panel*) in the anterior pituitary of female rat. The merged microscopic image showing colocalization (yellow). The experiments (**d-e**) are performed three times with similar observations.

The online version of this article includes the following source data and figure supplement(s) for figure 1:

**Source data 1.** TANGO β-Aggregation propensity.

**Source data 2.** Protein/peptide hormone sequence used in this study.

**Figure supplement 1.** ThioS co-localization of PRL and GAL expressing cells in female rat pituitary tissue.

**Figure supplement 2.** Double immunofluorescence of PRL-GAL in male rat pituitary tissue.

---

To explore these possibilities, we asked whether two hormones with completely different sequences, length and without any sequence similarity/identity (*Figure 2—figure supplement 1*) can co-assemble to form hybrid fibrils. To do this, we mixed the 1:1 molar ratio of both the hormones, incubated at 37 °C, and probed the amyloid formation by Thioflavin T (ThT) binding and CD spectroscopy for 15 days (*Figure 2b*, *Figure 2—figure supplement 1*). As a control, both hormones were incubated alone. Since PRL and GAL are known to form amyloid in presence of specific glycosaminoglycans such as chondroitin sulfate A (CSA) and Heparin (Hep) (*Maji et al., 2009a*), we also incubated both the hormones in presence of their respective glycosaminoglycans as positive controls. As expected, both PRL and GAL showed fibril formation in the presence of CSA and Hep, respectively. No ThT binding was observed by either PRL or GAL in absence of glycosaminoglycans (*Figure 2b*, *Figure 2—figure supplement 1*).

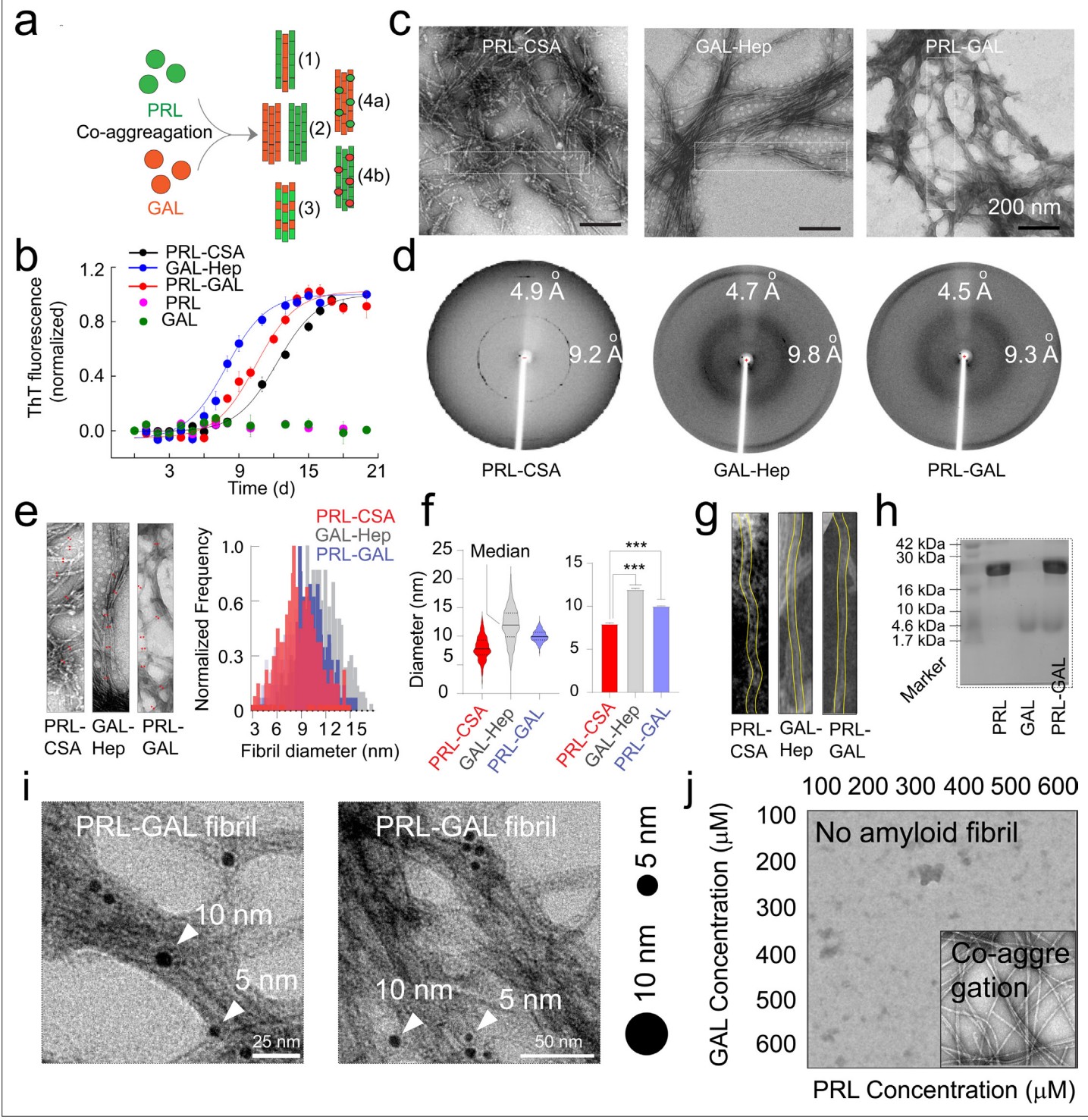

**Figure 2.** Amyloid aggregation kinetics & co-fibril formation by PRL and GAL. (**a**) Schematic showing different possibilities for the formation of PRL-GAL co-fibrils. (**b**) Normalized ThT fluorescence intensity over time showing faster aggregation kinetics for GAL-Hep followed by PRL-GAL and PRL-CSA. The experiment is performed three times with similar results. Values represent mean ± SEM. (**c**) TEM images showing amyloid fibrils for PRL-CSA, GAL-Hep, and PRL-GAL after 15 days of incubation. Representative images are shown. The dotted box marks are the representative area from which the fibril diameters are measured. (**d**) XRD of PRL-CSA, GAL-Hep, and PRL-GAL fibrils at day 15 showing ~4.7 Å meridional and ~10 Å equatorial reflections, as commonly seen for most amyloid fibrils (***Sunde et al., 1997***; ***Sunde and Blake, 1997***). (**e**) (*Left panel*) Representative TEM images showing fibril diameter measured at random positions (marked with red arrows) on individual fibrils. (*Right panel*) The normalized frequency distribution of fibril diameters of PRL-CSA, GAL-Hep, and PRL-GAL fibrils is shown. 200 data points are collected for individual samples for n = 3 independent experiments.

*Figure 2 continued on next page*

*Figure 2 continued*

(**f**) (*Left panel*) Median values of different fibril diameters are shown with violin plots. (*Right panel*) Average values of different fibril diameters are shown. Values represent mean ± SD. The statistical significance (***p ≤ 0.001, **p ≤ 0.01) is calculated by one-way ANOVA followed by an SNK post hoc test with a 95% confidence interval. (**g**) Representative TEM images of PRL-CSA, GAL-Hep, and PRL-GAL fibrils (scale bar-200 nm). From a single fibril, 200 data points are collected along the length to calculate the diameter. (**h**) SDS-PAGE depicting two bands for isolated aggregates from the PRL-GAL mixture (lane 3). The two bands correspond to PRL and GAL, which suggests that the isolated aggregates are composed of both PRL and GAL. (**i**) Amyloid fibrils obtained from the PRL-GAL mixture showing 10 nm gold particles (against GAL primary) and 5 nm gold particles (against PRL primary), confirming synergistic co-fibril formation by PRL and GAL (*Left and Right panel*). The experiment is performed three times with similar observations. (**j**) Schematic representation of incubation of PRL and GAL at various concentrations showing optimum concentration is required to initiate PRL-GAL co-aggregation.

The online version of this article includes the following source data and figure supplement(s) for figure 2:

**Figure supplement 1—source data 1.** Conformational transition and aggregation by PRL-GAL.

**Figure supplement 2—source data 1.** Congo red binding of hormone amyloids.

**Figure supplement 3—source data 1.** Time-dependent amyloid fibril formation by hormones using FTIR study.

**Figure supplement 4—source data 1.** Concentration regime of PRL-GAL co-aggregation.

**Source data 1.** Amyloid aggregation kinetics & co-fibril formation by PRL and GAL.

**Figure supplement 1.** Conformational transition and aggregation by PRL and GAL.

**Figure supplement 2.** Congo red (CR) binding of hormone amyloids.

**Figure supplement 3.** Time-dependent amyloid fibril formation by hormones using TEM and FTIR study.

**Figure supplement 4.** Concentration regime of PRL-GAL co-aggregation.

Strikingly, the mixture of both the hormones showed high ThT binding during time indicating amyloid formation (*Figure 2b*, *Figure 2—figure supplement 1*). CD spectroscopy showed a substantial decrease of helical content in PRL-GAL and PRL-CSA during aggregation. Notably, PRL is known to form amyloid without structural conversion to β-sheet (*Maji et al., 2009a*). On the other hand, GAL-Hep showed random coil to β-sheet conversion during amyloid formation (*Figure 2—figure supplement 1*). The lag times of aggregation for PRL-CSA, PRL-GAL, and GAL-Hep were calculated as ~9.2 days, ~ 8.8 days, and ~7 days, respectively (*Figure 2—figure supplement 1*). This is also consistent with the plot of molar ellipticity value (θ) at 222/218 nm with incubation time demonstrating the structural changes for each of the PRL/GAL samples (*Figure 2—figure supplement 1*).

Moreover, PRL-GAL aggregates showed fibril like morphology under TEM (*Figure 2c*), strong apple-green/golden birefringence under cross-polarized light (*Figure 2—figure supplement 2*), exhibited cross β-sheet diffraction patterns (*Sunde et al., 1997*; *Sunde and Blake, 1997*) ( ~ 4.7 Å for inter-strand and ~9.8 Å for inter-sheet) (*Figure 2d*) and showed FTIR peaks corresponding to β-sheet structure (*Jackson and Mantsch, 1995*; *Kong and Yu, 2007*; *Figure 2—figure supplement 3*). Similar observations supporting amyloid structure were also obtained for PRL-CSA and GAL-Hep aggregates (*Figure 2c–d* and *Figure 2—figure supplements 1–3*).

Although there are various possibilities regarding how PRL-GAL can form fibrils (any of the component hormones can form fibrils separately or together, as described in *Figure 2a*), we analyzed the frequency distribution of the fibril diameters from TEM images. Fibril diameters were measured from a particular sample (n = 200 randomized points) (*Figure 2e*, *left panel*) and the normalized frequency distribution was plotted (*Figure 2e*, *right panel*). Our data indicated that the diameter of PRL-CSA fibrils was least (*Figure 2e*, *right panel*) having a median value of ~7.5 nm (*Figure 2f*, *left panel*) and an average of ~7.8 nm (*Figure 2f*, *right panel*). The diameter of GAL-Hep fibrils was highest (*Figure 2e*, *right panel*) having a median value of ~12.3 nm (*Figure 2f*, *left panel*) and an average of ~12 nm (*Figure 2f*, *right panel*). Intriguingly, the PRL-GAL fibrils showed an intermediate diameter (median ~10 nm, average ~9.8 nm) (*Figure 2e–f*). This data suggest that PRL-GAL fibrils might be a new type of hybrid fibrils, which is neither similar to GAL nor PRL fibrils. We further analyzed the diameter from the same fibril bundle along its length. Consistent with our random point analysis (*Figure 2e–f*), we found that the diameter of a single PRL-CSA fibril was indeed the lowest (average ~7 nm) followed by PRL-GAL (average ~10 nm) and GAL-Hep (average ~12 nm) fibril (*Figure 2g*, *Figure 2—figure supplement 1*). Our data indicate that PRL-GAL fibrils possess unique morphology, which could be due to the incorporation of both PRL as well as GAL molecules into the same fibril (co-fibril).

To further analyze whether both PRL and GAL are part of the same insoluble fibril fraction, we isolated the fibrils using centrifugation and performed SDS-PAGE. The presence of two bands corresponding to PRL and GAL indicated that both PRL and GAL formed fibrils when co-incubated together (*Figure 2h*). Further, we performed immuno-electron microscopy (Immuno EM) with the PRL-GAL fibrils in the presence of secondary antibodies against PRL and GAL attached with 5 nm and 10 nm gold nanoparticles, respectively (*Figure 2i*). Co-localization of both 5 and 10 nm nanoparticles within the same fibril bundle confirmed that both PRL and GAL were part of the same fibril bundle (*Figure 2i*). However, due to extensive bundling of fibrils, it remains to be seen whether PRL and GAL are incorporated in the same filament or not.

Since 400 µM (each) PRL-GAL could undergo aggregation even in the absence of any helper molecules, we wanted to understand the optimum concentration and stoichiometry for PRL-GAL co-aggregation. To do this, we chose increasing concentration of PRL and GAL in an orthogonal (X-Y axis) manner (from 100 to 600 µM, each) (*Figure 2j*, *Figure 2—figure supplement 4*). All combinations of this concentration regime of hormone mixture were incubated for 15 days. We observed that amyloid fibril formation only occurred (within a feasible experimental timeframe [15 days]) as determined by ThT binding and electron microscope study (*Figure 2j*, *Figure 2—figure supplement 4*) when the concentration of each hormone PRL and/or GAL was kept ≥400 µM. The equal molar ratio of PRL: GAL and above 400 µM concentration of each hormone showed fibril formation. However, if any of the components (PRL or GAL) was taken below 400 µM, we could not observe fibril formation. Therefore, the area of the fibril space would be at a 1:1 ratio with little deviation on either side provided that the minimum concentration of each of PRL and GAL is 400 µM.

## Cross-seeding of PRL and GAL

Amyloid aggregation is a nucleation-dependent polymerization process (*Wood et al., 1999*; *Srivastava et al., 2019*). It is well-known fact that the presence of preformed nuclei of amyloid (also called 'seeds') greatly affects the kinetics of aggregation of monomers (*Daskalov et al., 2021*; *Ren et al., 2019*; *Morales et al., 2013*). Since PRL and GAL co-aggregate into mixed amyloids, we asked whether both PRL and GAL could cross-seed (*Daskalov et al., 2021*; *Ren et al., 2019*; *Morales et al., 2013*; *Ivanova et al., 2021*; *Hartman, 2013*) to induce amyloid aggregation of each other and help their possible storage in SGs. To understand this in detail, we performed both homotypic (PRL monomer+ PRL seed; GAL monomer+ GAL seed) and heterotypic seeding (PRL monomer+ GAL seed and GAL monomer+ PRL seed).

The preformed fibrils were sonicated to obtain PRL and GAL amyloid fibril seeds (see materials and methods). One percent, 2%, and 5% (v/v) of PRL and GAL seeds were mixed with freshly prepared 400 µM PRL and GAL, respectively, and incubated with slight agitation at 37 °C for homotypic seeding (*Figure 3b*, *Figure 3—figure supplement 1*). Our ThT fluorescence data showed accelerated aggregation by both PRL and GAL for homotypic seeding as lag time decreased significantly in the presence of 2% and 5% seeds (*Figure 3b and e*). However, 1% seed did not show any fibril formation for both PRL as well as GAL (*Figure 3b*, *Figure 3—figure supplement 1*) even after 10 days of incubation. Fibril formation via homotypic seeding mechanism was further supported by TEM imaging and time-dependent CD spectroscopic measurements (*Figure 3b–c*, *Figure 3—figure supplement 1*).

Similar experiments were done where various concentrations of PRL seeds were mixed with GAL monomer and GAL seeds were mixed with PRL monomer (heterotypic seeding). We hypothesized two possibilities of secondary nucleation (*Linse, 2017*)—the heterotypic monomers can be recruited at the ends of the seeds and facilitate the growth of amyloid fibril (elongation) or the seed surface will help in the nucleation of the heterotypic monomer (*Linse, 2017*; *Figure 3a*). Our data showed that GAL aggregation was accelerated with PRL seeds in a concentration-dependent manner (*Figure 3d*, *left panel*). Surprisingly, we observed no ThT fluorescence for all seed concentrations even after 10 days of incubation when GAL seeds were incubated with PRL monomers, indicating that GAL fibrils are incapable of inducing amyloid fibril formation of PRL monomers (*Figure 3d*, *right panel*). This was also evident with CD spectroscopic measurements (*Figure 3—figure supplement 2*). Interestingly, no aggregation was observed when PRL/GAL monomer was incubated in presence of different percentages of PRL-GAL mixed fibril seeds (1%, 2%, and 5%) as confirmed by CD spectroscopy, ThT fluorescence, and TEM imaging (*Figure 3—figure supplements 3–4*). This suggests that seeding (both homo and hetero) event is very specific for the life cycle of PRL/GAL amyloid formation

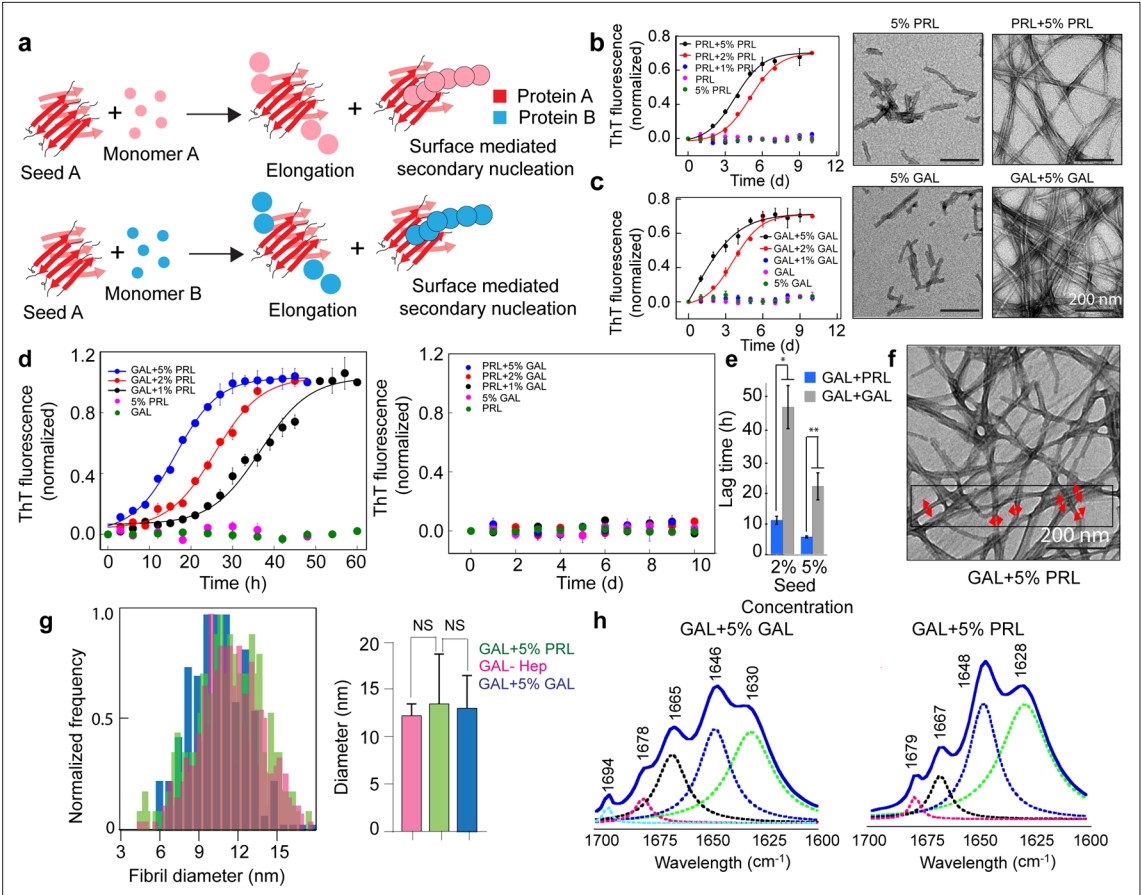

**Figure 3.** Seeding and cross-seeding of PRL and GAL. (**a**) Schematic showing possible homo and hetero seeding with fibril elongation and surface-mediated secondary nucleation mechanism for seed-mediated fibril growth. (**b–c**) Homo-seeding of PRL and GAL. (*Left panel*) PRL and GAL homo-seeding. Normalized ThT fluorescence intensity values with time indicating aggregation of PRL/GAL and in the presence of different concentrations of PRL seeds and GAL seeds respectively (2% and 5% v/v). Only seeds and only PRL/GAL was used as controls. (*Right panel*) The corresponding EM images of PRL/GAL seeds alone and PRL/GAL monomer in presence of 5% PRL/GAL seeds showing fibrils formation by PRL/GAL homo-seeding. (**d**) Cross-seeding of PRL and GAL. (*Left panel*) Normalized ThT fluorescence intensity values with time indicating aggregation of GAL in the presence of different concentrations of PRL seeds (1%, 2%, and 5% v/v). However, PRL in presence of different percentages of GAL seeds does not show any aggregation (*Right panel*). Only seed and only GAL/PRL were used as controls where no aggregation is observed. (**e**) The lag times of GAL aggregation in presence of 2% and 5% (v/v) PRL seeds and GAL seeds are compared. The values represent mean ± SEM. The significance (***p ≤ 0.001) is calculated using one-way ANOVA followed by an SNK post hoc test with a 95% confidence interval. (**f**) TEM images of GAL fibrils formed in presence of PRL seeds are shown. GAL fibrils formed in presence of 5% (v/v) PRL seeds are analyzed for frequency distribution (red arrows indicating the diameter of the fibrils measured for analysis). (**g**) (*Left panel*) Normalized frequency distribution of fibril diameter showing GAL fibrils formed in presence of PRL seeds have a similar diameter to GAL-Hep fibrils. A total of 200 random data points from different individual fibrils were collected from n = 3 independent experiments for the frequency distribution analysis. (*Right panel*) Average values of different fibril diameters are shown. Values represent mean ± SD. The statistical significance (***p ≤ 0.001, **p ≤ 0.01) is calculated by one-way ANOVA followed by an SNK post hoc test with a 95% confidence interval. (**h**) FTIR spectra showing fibrils of GAL +5% GAL seed and GAL +5% PRL seed are of similar secondary structure.

The online version of this article includes the following source data and figure supplement(s) for figure 3:

**Source data 1.** Seeding and cross-seeding of PRL and GAL.

**Figure supplement 1.** Aggregation and secondary structural transformations during seeding of PRL and GAL fibrils.

**Figure supplement 1—source data 1.** Aggregation and secondary structural transformations during seeding of PRL and GAL fibrils.

**Figure supplement 2.** Secondary structural transformations due to cross-seeding of PRL and GAL fibrils.

**Figure supplement 2—source data 1.** Secondary structural transformations due to cross-seeding of PRL and GAL fibrils.

**Figure supplement 3.** Cross-seeding of PRL and GAL by PRL-GAL co-fibrils.

**Figure supplement 3—source data 1.** Cross-seeding of PRL and GAL by PRL-GAL co-fibrils.

**Figure supplement 4.** Electron microscopy of cross seeding of PRL and GAL by PRL-GAL co-fibril seed.

*Figure 3 continued on next page*

*Figure 3 continued*

**Figure supplement 5.** Fitting of PRL-GAL cross-seeding kinetics. (**a, b, and c**).

**Figure supplement 5—source data 1.** Fitting of PRL-GAL cross-seeding kinetics.

in SGs. Important to note that these homotypic and heterotypic seeding studies were done in the absence of any helper glycosaminoglycans molecules. When lag times were compared, the heterotypic seeding rate was significantly higher than homotypic seeding for GAL aggregation suggesting surface-mediated secondary nucleation might be triggering GAL aggregation in the presence of PRL seeds (*Koloteva-Levine et al., 2020*; *Figure 3e*).

To further understand whether PRL seeds engage the GAL monomer for secondary nucleation (*Buell, 2019*) where elongation and/or surface-mediated aggregation could happen (*Linse, 2017*), fibril diameters of GAL (formed by PRL cross-seeding) were analyzed from the TEM images (*Figure 3f*). If only the elongation mechanism is occurring, GAL will form PRL-like fibrils. If PRL seeds engage GAL monomer for surface-mediated secondary nucleation, GAL will essentially form GAL-like fibrils. Intriguingly, analysis of frequency distribution of the GAL fibril diameters indicated that the distribution of diameter of PRL seeded GAL fibrils was very similar to that of GAL-Hep fibrils (*Figure 3g*, *left panel*). The median and mean fibril diameter analysis also suggests that GAL fibril formed in presence of PRL seed (median and mean diameter 12.35 nm) is very similar compared to GAL fibrils formed in presence of GAL seed (median and mean diameter 12 nm) as well as GAL fibrils formed in presence of heparin (median and mean diameter 11.20 nm) (*Figure 3g*, *right panel*, *Figure 3—figure supplement 1*). This means that the incorporation of GAL monomers does not happen to the PRL fibril end (elongation) (*Katzman and Saitoh, 1991*; *Ritter et al., 2005*; *Ren et al., 2019*; *Sarell et al., 2013*; *Cohen et al., 2018*), rather, GAL monomers use PRL seeds as surface and form amyloid fibrils via secondary (or surface-mediated) nucleation (*Törnquist et al., 2018*; *Ivanova et al., 2021*; *Hartman, 2013*; *Koloteva-Levine et al., 2020*) mechanism. This was further confirmed with the FTIR study, which suggested that GAL fibrils formed in presence of GAL seeds and GAL fibrils formed in the presence of PRL seeds possessed similar FTIR spectral signature, which was substantially different from the PRL-CSA fibril spectrum (*Figure 3h*, *Figure 2—figure supplement 3*).

We also performed the global fitting analysis of the PRL-GAL cross-seeding kinetics data at different PRL seed concentrations (1%, 2%, and 5%) using Amylofit (*Meisl et al., 2016*) (version 2.0), which has been extensively used to identify the underlying mechanism behind the kinetics of amyloid aggregation (*Kumari et al., 2021*; *Andreasen et al., 2019*; *Rasmussen et al., 2019*; *Frankel et al., 2019*). We observed the data poorly fit with the 'nucleation-elongation model', which exclusively considers primary nucleation and elongation steps. However, the kinetics data of GAL aggregation in presence of various PRL seeds (1%, 2%, and 5%) could be satisfactorily fit with the secondary nucleation model, which considers surface catalyzed secondary nucleation, along with classical elongation. Further, the mean residual error (M.R.E.) in fitting in the elongation mechanism was observed to be higher compared to the fitting using the model with surface catalyzed secondary nucleation. The fitting data, therefore, is in line with our hypothesis of surface-mediated amyloid assembly playing a dominant role in the aggregation mechanism of GAL in the presence of PRL seed (*Figure 3—figure supplement 5*).

## Specific interactions of PRL and GAL leading to amyloid aggregation

Next, we wanted to further investigate if interactions leading to co-aggregation and amyloid formation of PRL and GAL are specific to themselves. To do this, we co-incubated PRL with adrenocorticotropic hormone (ACTH) as this hormone is of similar length to GAL and does not form amyloid by itself (*Maji et al., 2009a*; *Ranganathan et al., 2012*). Similarly, GAL was also incubated with growth hormone (GH), a hormone structurally and functionally related to PRL (*Schmidt et al., 1991*; *Nilsson et al., 2001*; ). ThT aggregation kinetics and CD spectroscopy were performed at the beginning of the aggregation (day 0) and after 15 days of incubation for both PRL-ACTH and GAL-GH. Our data showed negligible ThT fluorescence for both PRL-ACTH and GAL-GH even after 15 days (*Figure 4a*) suggesting no co-aggregation. This observation was consistent with no structural conversion observed in CD and TEM, where the PRL-ACTH and GH-GAL mixtures were devoid of any fibrils (*Figure 4b*, *Figure 4—figure supplement 1*). Overall, our observations suggest that interaction and co-aggregation/amyloid formation by PRL and GAL are specific and are mutually beneficial for the storage of these hormones in SGs. This specific co-aggregation of PRL and GAL could be due to their favorable

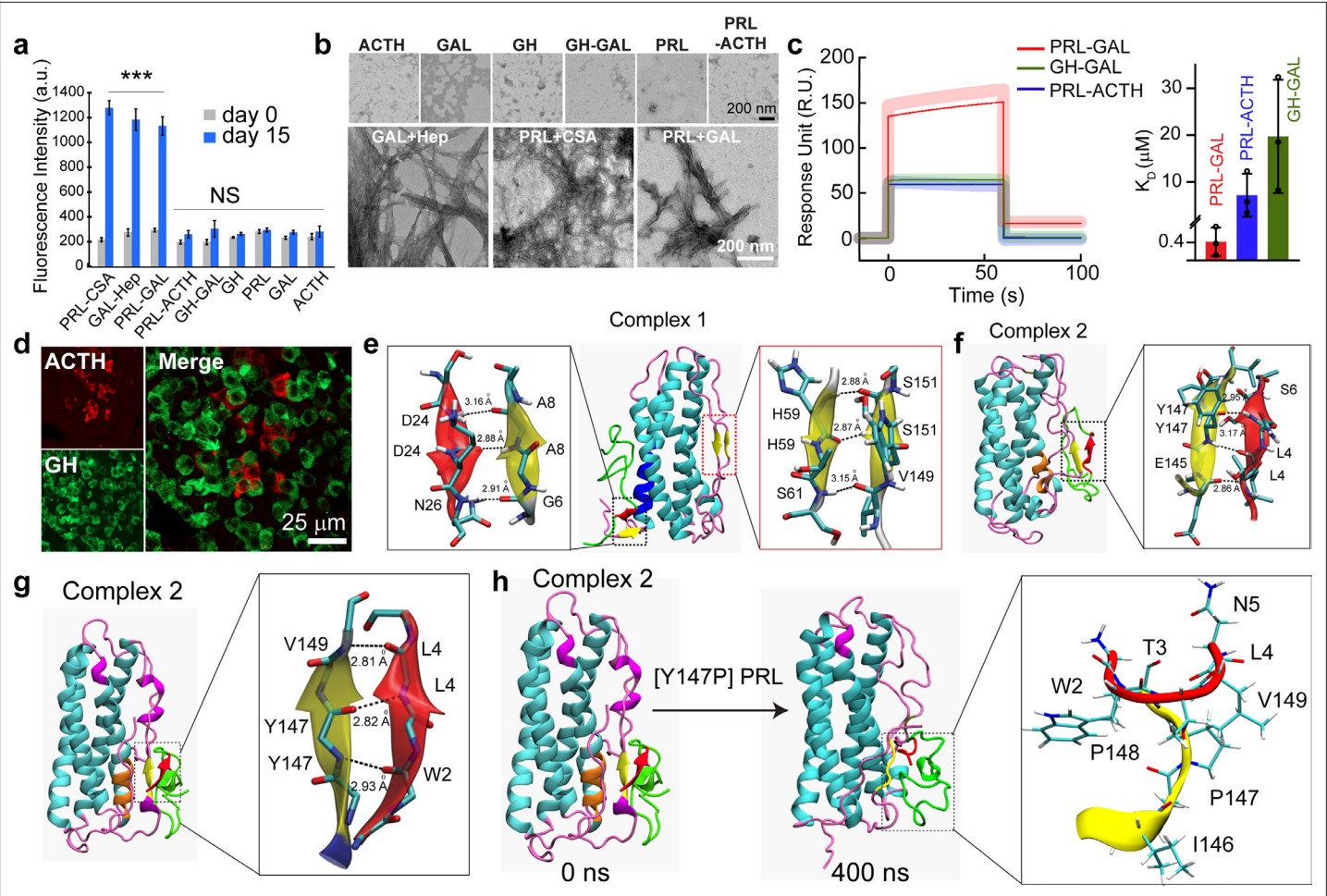

**Figure 4.** Specific interaction drives co-aggregation of PRL and GAL. (**a**) Comparative ThT fluorescence showing amyloid formation by different pairs of hormones at days 0 and 15. PRL-CSA, GAL-Hep, PRL-GAL showed the highest ThT fluorescence signals after 15 days of incubation. Values represent mean ± SEM for n = 3 independent experiments. The statistical significance is calculated between day 0 and day 15 for each sample using a t-test. (**b**) The morphology observed under TEM for various hormones and the mixture of hormone samples is shown (after 15 days of incubation). Amorphous structures are seen for PRL-ACTH and GAL-GH; whereas PRL-GAL, PRL-CSA, GAL-Hep showed fibrillar morphology similar to amyloids. The experiment is performed three times with similar observations. (**c**) (*Left panel*) Surface Plasmon Resonance (SPR) spectra showing strong binding of PRL on immobilized GAL compared to other pairs of hormones. (*Right panel*) The dissociation constant ($K_D$) of PRL to GAL showing strong interaction between PRL and GAL for their co-aggregation and co-storage. The experiments are performed three times with similar results. (**d**) Double immunofluorescence microscopic images of the anterior pituitary of female rats showing ACTH (red) and GH (green) expressing cells. The merged microscopic image (*right*) shows no co-localization of ACTH (red) and GH (green). The data indicate that ACTH and GH are not co-stored in the female rat anterior pituitary. The experiments were performed three times with similar observations. (**e**) Snapshot from in silico analysis (MD simulation) of PRL-GAL complex 1 using GROMOS 53a6 force field (when GAL is docked near residues 18–28 of PRL). (**f**) Snapshot showing MD simulation of PRL-GAL complex 2 (when GAL is docked near residues 80–88 of PRL) using GROMOS 53a6 force field. Complex 1 induced the formation of an antiparallel β-sheet at the PRL-GAL interface (6–8 PRL and 24–26 GAL) and also an intra-molecular parallel β-sheet in PRL itself (59–61 PRL and 149–151 PRL). Complex 2 shows the formation of a parallel β-sheet constituted by the β-strand from PRL and GAL (145–147 PRL and 4–6 GAL). (**g**) Snapshot of MD simulation of complex 2 using Amber ff99SB force field showing the appearance of parallel β-sheet at 147–149 residue of PRL and 2–4 residue of GAL. The snap-shot of complex 1 is included in *Figure 4—figure supplement 3*. (**h**) A point mutation is introduced in the PRL of the complex 2 structure, which is Y147P to examine if there is a loss in the β-sheet formation. The initial structure of the complex had the β-sheet formed between residues 147–149 of PRL and residues 2–4 of GAL (*Left panel*), which went missing during the 400 ns MD simulation run of the mutated system (*right panel*).

The online version of this article includes the following source data and figure supplement(s) for figure 4:

**Source data 1.** Specific interaction drives co-aggregation of PRL and GAL.

**Source data 2.** Parameter values for surface plasmon resonance spectroscopy (SPR) table.

**Source data 3.** Parameter values for surface plasmon resonance spectroscopy (SPR).

**Figure supplement 1.** Conformational transition by different pairs of hormone co-aggregation using CD.

*Figure 4 continued on next page*

*Figure 4 continued*

**Figure supplement 1—source data 1.** Conformational transition by different pairs of hormone co-aggregation using CD.

**Figure supplement 2.** Surface plasmon resonance spectroscopy (SPR) for inter-hormone interactions.

**Figure supplement 2—source data 1.** Surface plasmon resonance spectroscopy (SPR) for inter-hormone interactions.

**Figure supplement 3.** Protein-protein docking and MD simulations of PRL-GAL complexes.

**Figure supplement 4.** Mutation/deletion of certain interface residues disrupt PRL-GAL interaction.

interaction when the monomeric hormone is mixed together. This is further evident from the surface plasmon resonance (SPR) study, where GAL monomers were immobilized on a CM-5 chip and a range of concentrations of PRL monomeric protein was passed over it. We observed a significant increase in the response unit (RU) indicating binding of PRL to GAL (*Figure 4c*, *Figure 4—figure supplement 2*). The relative dissociation constant ($K_D$) was calculated to be $4.1 \times 10^{-7}$ M, which indicates the binding of PRL with GAL (*Figure 4c*). In comparison, we observed no significant binding when ACTH was passed through immobilized PRL or when GH was passed through immobilized GAL, which could be due to low but non-specific transient interactions between GH-GAL and PRL-ACTH (*Figure 4c*). This suggests that PRL monomers can readily bind GAL monomers, possibly contributing to their initial interaction that eventually drives the synergistic aggregation and amyloid formation. To further experimentally verify the specificity of PRL-GAL interaction, which is responsible for their co-aggregation and co-storage, we chose another combination of hormones (GH and ACTH), which generally expressed in anterior pituitary for their possible co-localization in the SGs of the female rat. Using the double immunofluorescence study, we found that GH and ACTH do not co-localize with each other (*Figure 4d*). The data suggest that specific interaction between two hormones might dictate their co-aggregation and co-storage irrespective of several hormones, which are present inside the anterior pituitary of the female rat.

To understand the mechanism of PRL and GAL interactions at the atomic level, docking, and molecular dynamics (MD) simulation studies were performed. GAL was docked at two different regions of PRL that showed high TANGO score (*Figure 1b and c*). Thus, two sets of PRL-GAL complexes were generated (*a*) *Set 1:* GAL was docked near residues 18–28 of PRL and *(b) Set 2:* GAL was docked near residues 80–88 of PRL. The lowest energy docked complexes from each set, named complex 1 and complex 2 respectively, were obtained (*Figure 4—figure supplement 3*). Both these complexes were then subjected to independent 250 ns long MD simulations to examine their stability. Since the choice of force fields may play an important role in the MD simulation results (*Cino et al., 2012*), we performed MD simulations with two different force fields – GROMOS 53a6 force field (*Oostenbrink et al., 2004*) and Amber ff99SB force field (*Tian et al., 2020*). As controls, we have also simulated individual PRL and GAL proteins. We observed that the PRL and GAL alone did not show any noticeable structural changes (*Figure 4—figure supplement 3*) during simulation time.

On the contrary, the PRL-GAL complexes exhibited significant conformational changes upon binding to each other, in both complexes (*Figure 4e–g*). The respective structures of complex 1 and complex 2 at the end of the MD simulation using GROMOS force field showed the structural transition from unstructured region to β-strand in both PRL and GAL with the emergence of a parallel or antiparallel β-sheet at the protein-protein interface (*Figure 4e and f*). The interaction of GAL in complex 1 induced the formation of an antiparallel β-sheet at the PRL-GAL interface (residues 6–8 of PRL, residues 24–26 in GAL) and also an intra-molecular parallel β-sheet in PRL itself (PRL residues 149–151 and 58–60) (*Figure 4e*). In complex 2, GAL induced the formation of parallel β-sheet at PRL-GAL interface (residues 145–147 of PRL, residues 4–6 of GAL) (*Figure 4f*). From the MD simulations using the Amber force field, however, the complex 1 did not show any notable structural changes, except that the terminal loops in PRL wrap around the existing secondary structures for higher stability (*Figure 4—figure supplement 3*). However, the PRL-GAL interactions in complex 2 resemble very well with the results from the GROMOS force field exhibiting a parallel β-sheet constituted by the β-strand from PRL and GAL proteins (*Figure 4g*). These results corroborate very well with our experimental data that suggested the formation of amyloids when PRL and GAL were co-aggregated. The residues involved in the formation of this β-sheet in complex 2 were PRL residues 147–149 and GAL residues 2–4 in the Amber ff99SB force field (*Figure 4g*). Thus, irrespective of the force field used, our MD simulation

results convincingly show that the co-aggregation of PRL and GAL induce the formation of β-sheet at the protein interface.

To validate our MD simulation results, further we have performed mutagenesis studies by mutating or deleting certain PRL/GAL residues in silico. We have examined four different cases where GAL residues 2–5 (WTLN) (case 1); PRL residues 146–149 (IYPV) (case 2) are mutated to alanine (*Figure 4— figure supplement 4*). These amino acids were observed to involve in the inter-protein β-sheet formation.

In the other two cases, we introduced a point mutation in PRL as Y147P, since proline is known as β-sheet breaker (*Soto et al., 1998*) (case 3), and in case 4 we deleted first two N-terminal residues (GW) of GAL (*Figure 4h* and *Figure 4—figure supplement 4*). We did not simulate the PRL deletion system, as deletion at the middle of the PRL helical structure (PRL residues 146–149) would bring in conformational changes in the PRL secondary structure itself, and correlating that with our interest of PRL-GAL β-sheet formation will be difficult.

We further simulate the systems for 400 ns (150 ns for two residues deletion in GAL) after introducing these changes. Interestingly, in all the cases (Case 1 to Case 4), we observed loss of interactions (H-bonds) at the PRL-GAL interface, which eventually led to the loss of β-sheet structures (*Figure 4h*, *Figure 4—figure supplement 4*). It is worth mentioning here that a single mutation in PRL (Y147P; Case 3) was sufficient to disrupt the secondary structure formation, suggesting its key role in inducing the formation of β-sheet during co-aggregation of PRL and GAL (*Figure 4h*). Thus, the MD simulation results convincingly show that mutation or deletion at the PRL-GAL interface could result in the loss of β-sheet formation, which eventually affects the PRL-GAL co-aggregation.

## Release of functional PRL and GAL from PRL-GAL amyloids

Protein/peptide misfolding and aggregation lead to irreversible amyloid formation, which is stable and does not readily disassemble to monomer. However, many studies recently showed the release of monomers and oligomers from disease-associated amyloids (*Bemporad and Chiti, 2012*; *Cascella*

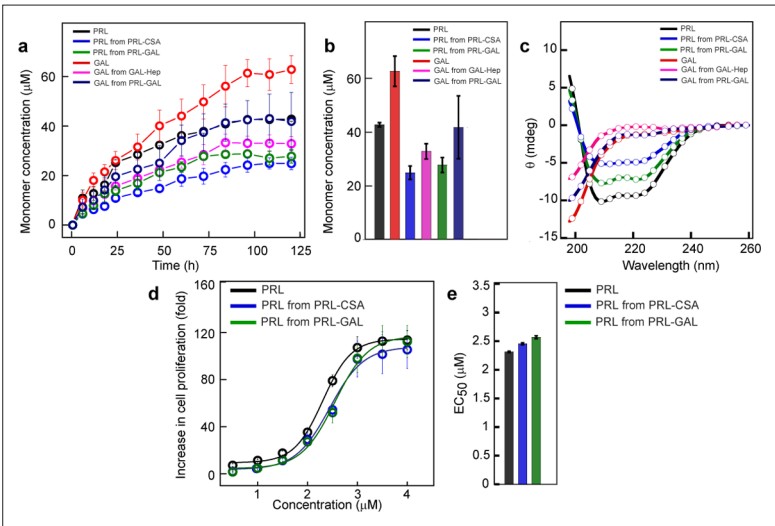

**Figure 5.** Monomer release from PRL and GAL amyloid. (**a**) The kinetics of monomer release from various amyloids showing the continuous release of monomeric hormones. The experiment is performed three times with similar results. Values represent mean ± SEM for n = 3 independent experiments. (**b**) Saturation concentrations of different released monomers from fibrils along with the monomeric controls are shown. Values represent mean ± SEM for n = 3 independent experiments. (**c**) The secondary structure of released monomers showing their corresponding native secondary structures as confirmed by the CD. (**d**) Nb2 cell proliferation study showing biological activity of released PRL from either PRL-CSA or PRL-GAL fibrils. Freshly dissolved protein was used as a control. Values represent mean ± SEM for n = 3 independent experiments. (**e**) EC$_{50}$ values showing the released PRL monomers have similar bioactivity compared to freshly dissolved monomeric PRL. Values represent mean ± SEM for n = 3 independent experiments.

The online version of this article includes the following source data for figure 5:

**Source data 1.** Monomer release from PRL and GAL amyloid.

*et al., 2021*). In contrast, amyloid formation related to SGs biogenesis is reversible and should be able to release functional monomers in the extracellular space for their function (*Maji et al., 2009a*; *Jacob et al., 2016*; *Anoop et al., 2014*). To address whether any functional advantage of co-aggregation over homotypic aggregation by PRL and GAL, we determined the relative monomer release capability of PRL/GAL monomer, and PRL-GAL co-amyloids; along with preformed fibrils of PRL (in the presence of CSA) and GAL (in the presence of Hep) using dialysis method (*Maji et al., 2009a*; *Jacob et al., 2016*; *Anoop et al., 2014*). The concentration of released monomers (if any) in the dialysate was measured by UV-Vis spectroscopy at different time points. Intriguingly, we found that PRL-CSA, GAL-Hep, and PRL-GAL amyloids could indeed release monomers with time (*Figure 5a*). Interestingly, the amyloid fibrils of PRL and GAL formed in presence of glycosaminoglycans (CSA and Hep, respectively) released monomeric hormones in a slow and sustained manner upon dilution in 10 mM Tris-HCl, pH 7.4 (*Figure 5a*).

However, the release of monomeric PRL and GAL hormones from the co-aggregated PRL-GAL fibril was faster compared to the PRL and GAL released from their glycosaminoglycans-mediated fibrils. This was confirmed by their release profile as well as saturation concentrations (*Figure 5a–b*). CD spectroscopy of the dialysate showed that the released PRL monomer retained its native conformation (*Figure 5c*).

Next, we performed the cell proliferation assay with the released PRL monomer to check for the retention of bioactivity of PRL. We used the Nb2 cell line for this study (*Bulatov et al., 1996*). Nb2 cells are rat lymphoma cells with significant expression of PRL receptors on the cell surface (*Bulatov et al., 1996*; *Lebrun et al., 1994*). These cells require PRL for proliferation or mitogenesis (*Lebrun et al., 1994*; *Upadhyay et al., 2016*). We observed that released PRL monomers obtained from PRL-CSA fibrils and PRL-GAL fibrils were functional as they can induce cell proliferation in a dose/concentration-dependent manner (*Figure 5d*). The $EC_{50}$ values of the PRL released from PRL fibrils are similar to the freshly prepared PRL monomer, suggesting no difference in their functionality (*Figure 5e*).

## Discussion
### PRL-GAL co-storage is facilitated by their co-aggregation

Amyloids are ordered protein aggregates comprised of cross-β-sheet motifs where β-sheets are parallel, and individual β-strands are perpendicular to the fibril axis (*Sunde et al., 1997*; *Maji et al., 2009b*). Despite their association with diseases, amyloids are also known to be involved in the native functions of host organisms including mammals (*Chiti and Dobson, 2006*; *Fowler et al., 2007*). Interestingly, the synergistic amyloid formation through co-aggregation and cross seeding by heterologous proteins/peptides is also evident in the disease-associated proteins such as α-Synuclein-Tau (*Waxman and Giasson, 2011*; *Moussaud et al., 2014*), α-Synuclein-amyloid β (*Köppen et al., 2020*; *Bassil et al., 2020*). However, these mechanisms are still elusive for functional amyloids despite their relevance in hormone co-storage and co-release from SGs. Here, we explored the synergistic aggregation and amyloid formation by two human hormones PRL and GAL, which are highly relevant in SGs biogenesis. Although 23 kDa PRL (*Teilum et al., 2005*; *Keeler et al., 2003*) (mostly helical) has no resemblance of sequence, length, and structure with unstructured 3.1 kDa GAL (*Evans and Shine, 1991*; *Bersani et al., 1991*), it was reported that both of these hormones are co-stored in the lactotrophs of the anterior pituitary in the female rats (*Hyde et al., 1991*; *Steel et al., 1989*). Moreover, their release is also modulated by the same secretagogues (*Koshiyama et al., 1987*; *Murakami et al., 1993*; *Koshiyama et al., 1990b*). Co-storage and co-release of PRL and GAL suggest that these hormones might aggregate together to form amyloids within the same SGs. Our immunofluorescence study with female rat anterior pituitary tissue probed that not only both PRL and GAL are colocalized together; they are indeed in the amyloid form as suggested by OC and ThioS staining (*Figure 1* and *Figure 1—figure supplement 1*). Interestingly when PRL and GAL were co-incubated with a 1:1 or higher molar ratio in vitro, they co-aggregated synergistically to form amyloid fibrils in conditions similar to their storage in SGs without the requirement of any helper molecules (*Reggio and Palade, 1978*; *Zanini et al., 1980*).

Co-aggregation can happen in a scenario where PRL and GAL can promote the aggregation of each other but exclusively form individual, homotypic fibrils (*Figure 2*). Further, one of them might have the conformational advantage to form amyloids and other proteins/peptides can simply adhere to it,

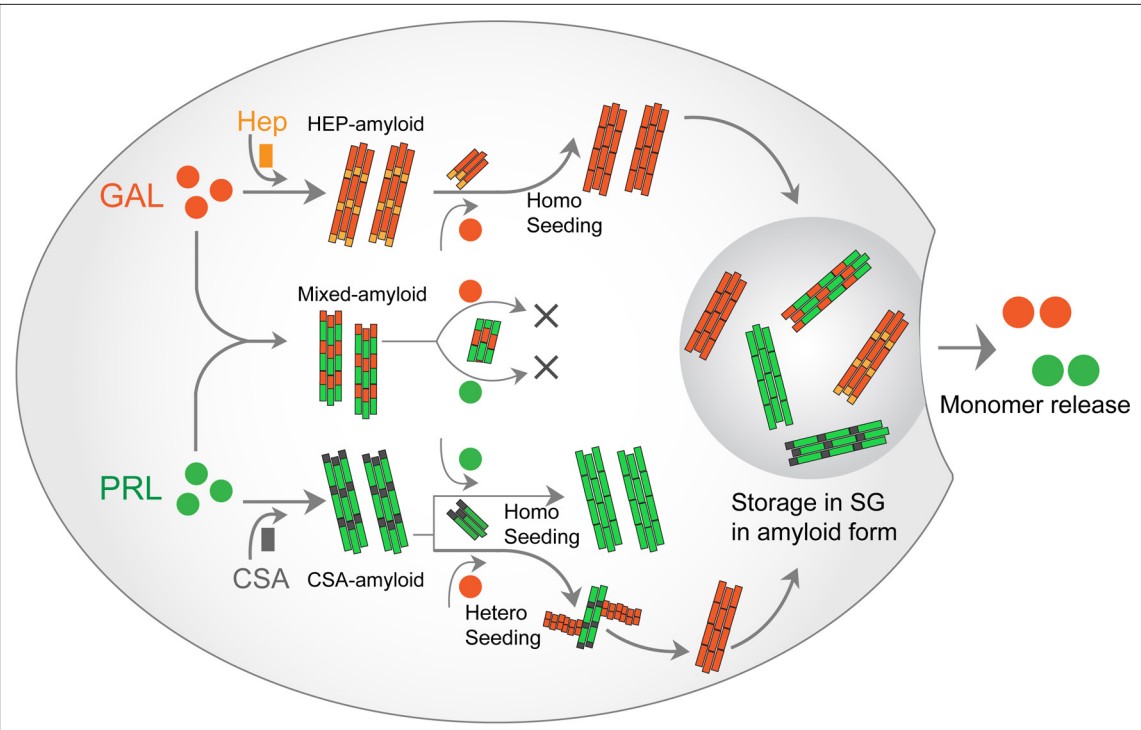

**Figure 6.** PRL-GAL homo and hetero amyloid life cycle for SGs. PRL and GAL form the amyloid fibrils in the presence of specific glycosaminoglycans (CSA and Hep, respectively), which can be auto-catalytically amplified by their respective seeding with preformed fibrils. This seeding however does not require any glycosaminoglycans. PRL-GAL also synergistically co-aggregate to form hybrid amyloid fibrils, which are not capable of seeding either to PRL or GAL. These amyloid fibril species can together or individually reconstitute the SGs of PRL-GA storage, which can release functional PRL and GAL into the extracellular space.

maintaining its monomeric structure (*Figure 2*). Another possibility is that PRL-GAL is incorporated within the same fibrils and/or filaments (*Figure 2*). Immuno-EM studies with secondary antibodies tagged with different-sized gold nanoparticles showed that both PRL and GAL are present within the same fibrils suggesting they might interact together to form hybrid, heterotypic fibrils (*Figure 2*). The incorporation of heterogenous proteins to make hybrid fibrils is also shown for α-Synuclein-Tau and for other proteins (*Köppen et al., 2020*; *Waxman and Giasson, 2011*; *Moussaud et al., 2014*; *Bassil et al., 2020*). The ability of co-aggregation without glycosaminoglycans (note that individual hormones are unable to form amyloids) suggests that in glycosaminoglycans deficient conditions in vitro, PRL and GAL will preferably form mixed amyloid for their storage. However, their co-aggregation does not preclude the possibility of individual storage of each hormone in the different SGs (mediated via glycosaminoglycans). Furthermore, individual hormone aggregation can be seeded with their respective fibrils seeds as shown for PRL and GAL even in absence of glycosaminoglycans (*Figure 6*). All these possibilities suggest that functional amyloid fibril formation might be very useful for host organism and therefore is produced in an autocatalytic manner inside the secretory cells.

## Unidirectional cross seeding of GAL by PRL fibril seeds: The possible facilitator of GAL storage in SG

Homologous seeding of amyloid aggregation is relatively more abundant in nature due to quick and feasible templating mechanism involving a single protein, which decreases the lag time for aggregation (*Eden et al., 2015*; *Arosio et al., 2015*). Although heterogeneous seeds can also provide template/surface for a different protein/peptide to initiate their aggregation (cross-seeding), this phenomenon is less abundant because of high thermodynamic and conformational barrier between the two different protein/peptides (*Daskalov et al., 2021*; *Ivanova et al., 2021*; *Hartman, 2013*). It was shown that sequence similarity between two amyloidogenic proteins is crucial for their cross seeding capability (*Krebs et al., 2004*). However, the ability of cross seeding also depends on the

conformation of the seed and its compatibility with the monomeric protein, which creates different cross seeding barrier (*Daskalov et al., 2021*; *Ren et al., 2019*; *Walker et al., 2013*) (also species barrier for prion diseases).

Our data showed PRL amyloid fibrils can cross-seed GAL monomers. Strikingly, GAL fibrils were not able to cross seed PRL monomers suggesting specificity and regulation in functional amyloid formation (*Figures 3 and 6*). Unidirectional, as well as bidirectional seeding between amyloid proteins, are indeed evident. Cross seeding between amyloid β–42 and hIAPP, where amyloid β–42 seeds can cross seed hIAPP to promote aggregation but hIAPP seed is unable to induce aggregation to amyloid β–42 monomers (*O'Nuallain et al., 2004*) (*unidirectional cross seeding*). In contrast, α-Synuclein and Tau seeds can accelerate each other's amyloid aggregation suggesting a bidirectional cross seeding mechanism (*Waxman and Giasson, 2011*; *Moussaud et al., 2014*).

Since PRL has less propensity to form amyloid due to higher structural stability, GAL seeds are unable to provide surface or structural compatibility for PRL cross seeding. In contrast, GAL is a highly disordered peptide, which might easily bind to the PRL amyloid surface nonspecifically and increase its stability, and proceeds the aggregation. This suggests that tight regulation of co-species aggregation ensures the right amount of storage of PRL and GAL in SGs with physiologically appropriate ratios. Furthermore, the amyloid-prone sequence of PRL could also be sequestered inside the helical structure, which might not be compatible with the surface and/or amyloid core structure of GAL to mediate the cross-seeding. Previously it was shown for K18 and K19 of Tau isoforms that K19 fibrils can cross seed K18 through the catalytic motif of R3, whereas K18 fibrils with the catalytic center as R2 is unable to seed K19 (*Yu et al., 2012*) (lacking R2 motif). The difference in seeding could also be due to the relative tendency of amyloid fibril formation by GAL and PRL. We propose that 'PRL seeding of GAL' is not due to elongation (which requires specific interaction) as PRL cross-β spine might not be accessible for GAL. In this context, we previously proposed that a small segment of GH (structurally similar to PRL) might be involved in fibril formation (*Jacob et al., 2016*) while other structural domains might surround this cross-β spine as proposed for RNaseA fibrils model (*Sambashivan et al., 2005*).

In contrast, GAL being a short neuropeptide has the conformational flexibility to adhere to the surface of the PRL amyloid seeds and subsequently can undergo surface-mediated secondary nucleation (*Andersen et al., 2009*). Afterwards, more and more GAL monomers will be accommodated on the PRL seed surface as the fibril growth progresses—resulting in similar bulk fibril morphology like GAL fibrils formed by homo-seeding (*Hartman, 2013*; *Koloteva-Levine et al., 2020*; *Figure 3*). The more efficient (faster kinetics) cross-seeding of GAL amyloid aggregation by PRL seeds compared to GAL seeds (homo-seeding) also indicates a possibility of the greater abundance of surface-mediated secondary nucleation mechanism for the former scenario (*Koloteva-Levine et al., 2020*; *Figure 3*).

We hypothesized that specific interaction between GAL and PRL synergistically facilitates their aggregation into amyloid. This is further evident from the SPR analysis showing strong interaction and direct binding between the monomeric forms of PRL and GAL in contrast to the other hormone pair, PRL-ACTH or GH-GAL (*Figure 4*). PRL-GAL interaction and conformational transition are further supported using in silico study, which showed interaction of the PRL N-terminal loop and GAL, where GAL promotes the conformational transition of the N-terminus of PRL into the β-sheet structure (*Figure 4*). The interaction of PRL and GAL is mandatory for their aggregation, as no amyloid fibril formation was observed when PRL and GAL were incubated alone.

## Functional implications of PRL-GAL co-aggregation and unidirectional cross-seeding

Our study probes that PRL-GAL co-aggregation is one of the key mechanisms for PRL and GAL storage inside the SGs and their subsequent release as functional hormones. PRL and GAL can also undergo homotypic aggregation in the presence of specific glycosaminoglycans like CSA or Hep, respectively. However, homotypic aggregation might not be sufficient for PRL/GAL protein homeostasis in the anterior pituitary especially during lactation in females, which could be achieved by PRL-GAL co-aggregation (*Figure 2*). It is known that PRL and GAL are co-stored in the lactotrophs of the female rat anterior pituitary (*Hyde et al., 1991*; *Steel et al., 1989*). Peptide hormone GAL is known to be highly expressed in the hypothalamus and performs various tropic activities including regulating the secretion of other neuropeptide hormones and neuronal differentiation (*Lundström et al., 2005*). Interestingly, GAL overexpression in the anterior pituitary during lactation is known to regulate PRL

expression, storage, and release (*Kaplan et al., 1988*). Storing a high amount of PRL in the anterior pituitary during lactation is also important for the development of lactotrophs (*Wynick et al., 1998*). This is evident as the loss of function mutation in the endogenous *GAL* gene in mice has shown a drastic decrease in PRL level compared to the wild-type control mice (*Wynick et al., 1998*). This was accompanied by reduced PRL secretion, failure in mammary gland maturation, and loss of lactotroph proliferation upon estrogen treatment in the mutant mice (*Wynick et al., 1998*). On the other hand, estrogen treatment enhanced the GAL mRNA by several orders of magnitude (*Wynick et al., 1993*), which was subsequently shown to be highly effective for lactotroph proliferation and subsequent PRL production and release (*Wynick et al., 1993*). However, functional implications due to the loss of PRL-synthesis/function for GAL storage and its release are yet to be established.

The present co-aggregation data clearly shows that GAL and PRL are interdependent on their storage within the same SGs in the amyloid-like state (without helper molecules), which facilitates the controlled and efficient release of these hormones. This is supported by the fact that functional monomer release from PRL-CSA amyloid and GAL-Hep amyloid is much slower compared to the functional monomer release from mixed amyloids of PRL-GAL (*Figure 5*). When GAL monomers release from the co-aggregates (PRL-GAL), it ensures further release of PRL and other tropic factors both in an autocrine and paracrine manner (*Cai et al., 1998*). The co-aggregation and unidirectional cross-seeding mechanism not only supports PRL storage and release but also helps to maintain a very low titre of unstable GAL to efficiently store in SGs for future use. However, a substantial decrease in physiological PRL levels would inevitably affect the storage of GAL and its subsequent release as they are in a positive feedback loop. On the other hand, over-production of GAL may result in hypersecretion of PRL—leading to prolactinoma (*Cai et al., 1999*). Therefore, tight regulation is required for GAL synthesis and secretion. Taken together, our findings probe that co-aggregation and co-storage would allow much higher efficiency for hormone release and function than their individual storage. The impairment of PRL-GAL co-aggregation would very likely be detrimental in achieving the PRL-mediated lactation and other tropic functions of GAL.

Further, as efficient storage and release of GAL are necessary for PRL homeostasis, GAL deficiency or functional mutation inhibits the PRL release (*Wynick et al., 1998*). GAL is also responsible for the regulation and secretion of other neuropeptides such as VIP (*Wynick et al., 1993*). Intriguingly, lactotrophs that do not produce GAL, are also highly sensitive to this hormone but not to the other secretagogues such as VIP peptide (*Wynick et al., 1993*), suggesting the essential requirement of GAL for several pituitary functions. In contrast to the requirement, only a minority of lactotrophs ( ~ 9% of cells) (*Wynick et al., 1993*) produce the GAL. Apart from the synthesis issue, its half-life (turnover cycle) is also reported to be low (3–4 min in circulation) (*Hinghofer-Szalkay et al., 2006*) in comparison to hormone like PRL ( ~ 41 min) (*Yoshida et al., 1991*; *Yu et al., 2019*), possibly due to its short length and unstructured nature (*Evans and Shine, 1991*; *Bersani et al., 1991*). Therefore, efficient storage of GAL is essential not only for PRL storage/release (which is required during pregnancy/lactation) but also for performing other pituitary functions (*Kask et al., 1997*). In this context, cross-seeding by PRL amyloids increases the chances of GAL storage and subsequent release. On the other hand, PRL storage needs tight regulation due to the risk of developing prolactinoma upon over-secretion. Interesting to note that templating mechanism (seeding) requires the client monomers to change their fold—which requires a substantial amount of energy. As discussed earlier, it is very likely that for a folded protein like PRL, it is thermodynamically less favorable to undergo a large conformational transition upon templating by GAL seeds. On the other hand, GAL is an unstructured neuropeptide and can easily be templated by PRL seeds. This could be an evolutionarily optimized design by nature so that tight regulation of GAL storage and functions can be regulated.

Since PRL and GAL cross-talk via a positive feedback loop for both their storage and release, a decrease in GAL level would also decrease the storage and release of PRL. This is also evident from our co-aggregation and unidirectional cross-seeding data. Exploiting this phenomenon, one could argue that by controlling (reducing) excess GAL, it is possible to prevent PRL hypersecretion (prolactinoma). This could, in principle, be achieved by targeting excess GAL in the bloodstream using various methods including specific antibodies, targeted proteolysis, or by engineering small molecules/metabolites capable of inhibiting GAL secretion/functions as previously demonstrated for various hormones (*Gera et al., 2020*; *Békés et al., 2022*; *Lu et al., 2019*; *Slastnikova et al., 2018*).

Further, instead of targeting GAL, other metabolites such as tryptophan amino acid can also be used that target PRL directly to control its release (*MacIndoe and Turkington, 1973*).

One more interesting aspect of functional amyloids is that they must exhibit sustained release of functional monomers at the target site. It will be of great interest in the future to design GAL analogs that can form stable co-amyloids with PRL, which would be incapable of releasing PRL monomers—thereby controlling the excess secretion of PRL. However, translating the current knowledge into therapeutics has challenges—primarily because, although PRL-GAL co-aggregation and unidirectional cross-seeding is very efficiently designed by nature, both hormones can also form functional amyloids independently in the presence of helper molecules such as glycosaminoglycans (*Figure 6*).

Overall, the current study demonstrates that contrary to disease-associated co-aggregation, which promotes the spread of the disease, co-aggregation of hormones is specific and functional in the SGs. Co-aggregation and amyloid formation of these structurally dissimilar hormones indicate the relevance of amyloid as a crucial aspect of cellular sorting and storage in SGs biogenesis.

# Materials and methods

## Key resources table

| Reagent type (species) or resource | Designation | Source or reference | Identifiers | Additional information |
|---|---|---|---|---|
| Cell line (*Rattus norvegicus*) | Nb2 cells (rat lymphoma cells) | PMID:27277580 Kind gift from Prof. Amulya K. Panda, National Institute of Immunology, New Delhi | | |
| Biological sample (*Rattus norvegicus*) | Rat pituitary tissue sections (Adult Sprague-Dawley rats) | NISER Bhubaneswar Institutional Animal Ethical Committee (IAEC) at NISER, Bhubaneswar | Protocol Numbers: NISER/SBS/ AH-210 and NISER/SBS/AH-212 | Approved by the Committee for the Purpose of Control and Supervision of Experiments for Animals (CPCSEA), New Delhi, India |
| Antibody | Anti-PRL (guinea pig polyclonal) | A.F. Parlow, National Hormone and Pituitary Program (NHPP) | AFP7192490 | 1: 1500 |
| Antibody | Anti-GAL (mouse polyclonal) | Abcam | Ab216399 | 1:1500 |
| Antibody | OC antibody (rabbit polyclonal) | Abcam | Ab126468 | 1:500 |
| Antibody | Goat anti-rabbit FITC (polyclonal) | Thermo Fisher Scientific, USA | 65–6111 | 1:500 |
| Antibody | Goat anti-mouse FITC (polyclonal) | Thermo Fisher Scientific, USA | 31,569 | 1:500 |
| Antibody | Goat anti-guinea pig Alexa Fluor 555 (polyclonal) | Thermo Fisher Scientific, USA | A21435 | 1:500 |
| Antibody | Anti-ACTH (rabbit polyclonal) | PMID:22403619 Kind gift from A.F. Parlow, National Hormone and Pituitary Program (NHPP) | | 1:1000 |
| Antibody | Anti-GH (goat polyclonal) | R&D Systems | AF1566 | 1:1000 |
| Antibody | Alexa Fluor-594 anti-rabbit (polyclonal) | Thermo Fisher Scientific, USA | A-21207 | 1:500 |
| Antibody | Goat anti-mouse Alexa fluor 555 (polyclonal) | Thermo Fisher Scientific, USA | A32727 | 1:500 |
| Antibody | Alexa fluor 488 anti goat (polyclonal) | Thermo Fisher Scientific, USA | A-11055 | 1:500 |
| Peptide, recombinant protein | Galanin | USV Limited (Mumbai, India) | Custom synthesis | |
| Software, algorithm | KaleidaGraph | | Version 4.0 | |

## Chemicals and reagents

The chemicals were obtained from Sigma Chemicals or other sources with the highest purity available.

## Expression and Purification of human prolactin (PRL)

PRL was expressed as per the protocol reported (*Sankoorikal et al., 2002*) with little modification. The human PRL plasmid was obtained as a kind gift from Prof. Dannies and Prof. Hodsdon from Yale University. BL21 (DE3) *E. coli* cells were transformed with the human PRL gene encoded in the pT7L plasmid and were made to grow in terrific broth (TB) followed by IPTG induction for 4 hr. After harvesting the cells at 8000 rpm for 20 min, the pellet was dispersed in 20 mM Tris-HCl, pH 8.0 (with added protease inhibitor cocktail, Roche). After that, the cells were sonicated (2 s on, 1 s off at 50% amplitude) for 20 min to complete cell-lysis, which was then centrifuged at 15,000 rpm for half an hour to recover the inclusion bodies (IB). The cell pellet containing IBs was subsequently washed two times with 0.5% triton-X and then it was dissolved in urea (8 M) with 2% (v/v) β-mercaptoethanol. The solution was then dialyzed against 20 mM Tris-HCl, pH 8.0 so that the PRL protein refolds to its native state. After dialysis, the protein solution was again centrifuged at 15,000 rpm for 1 hr and was loaded in an anion exchange column (Resource Q, GE healthcare) through an AKTA purifier FPLC system (Cytiva). The protein was eluted through NaCl (1 M) gradient and subsequently lyophilized after snap-freezing in liquid nitrogen. Size exclusion profile (SEC) suggested the protein is monomeric in nature and the purity of the protein is further checked by SDS-PAGE and MALDI-TOF spectrometry. CD spectroscopy was also performed to confirm that the purified PRL has refolded to its native helical conformation.

## Aggregation of PRL and GAL in the presence of glycosaminoglycans

PRL was dissolved in Milli-Q water and buffer exchanged to 20 mM phosphate buffer with 100 mM NaCl, pH 6.0, 0.01% sodium azide using 10 kDa mini dialysis units (Thermo Scientific Slide-A-Lyzer). 5 mM solution of CSA (Sigma, USA) was prepared in the same buffer, and appropriately mixed with PRL to obtain an ultimate concentration of 400 μM for both PRL and CSA. Similarly, for GAL aggregation in the presence of Hep, GAL peptide was dissolved in 20 mM phosphate buffer containing 100 mM NaCl, pH 6.0, 0.01% sodium azide to obtain a concentration of 500 μM. Hep solution from a stock of 5 mM (made in the same buffer) was then mixed with GAL solution to obtain an ultimate concentration of 400 μM for both GAL and Hep. These tubes containing PRL and GAL of various mixtures were kept into an Echo Thermmodel RT11 rotating mixture with a speed of 50 rpm for 15 days inside a 37 °C incubator. As a control, 400 μM PRL and GAL alone in the same buffer, was also incubated in a similar condition. The secondary structural transition was monitored by CD spectroscopy and amyloid formation by ThT binding assay at various time points. Finally, Congo red (CR) binding studies and TEM imaging was used to confirm amyloid fibril formation.

## Co-Aggregation study of PRL and GAL

For the co-aggregation study, PRL was dissolved in Milli-Q water and buffer exchanged to 20 mM phosphate buffer with 100 mM NaCl, pH 6.0, 0.01% sodium azide using 10 kDa mini dialysis units (final concentration was 800 μM) (Thermo Scientific Slide-A-Lyzer). GAL was also dissolved in the same buffer to obtain an 800 μM solution. After that, each of the solutions was mixed to obtain 400 μM of PRL-GAL mix and was incubated with slight agitation at 37 °C for 2 weeks. 400 μM of each PRL and GAL was incubated alone as controls. For co-aggregation of PRL and GAL with other hormones, separate solutions of PRL, GAL, GH, and ACTH were freshly dissolved and prepared in identical solution condition as above. Each solution was mixed to obtain 400 μM each of PRL-ACTH and GAL-GH mixture and was incubated with slight agitation at 37 °C for 2 weeks. A total of 400 μM of each of PRL, GAL, GH, and ACTH were also incubated alone as controls. The secondary structural transition and amyloid formation of the incubated solutions were monitored by CD spectroscopy and ThT-binding assay during various time points. After 15 days of incubation, the morphology of the incubated samples was analyzed by TEM.

## Circular dichroism spectroscopy (CD)

For CD measurement, protein/peptide aliquots were diluted in 20 mM phosphate buffer with 100 mM NaCl, pH 6.0, 0.01% sodium azide to 200 μl and the final concentration protein/peptide was 10 μM. CD spectra were taken using a JASCO 810 instrument where the sample was loaded in a quartz cell of 0.1 cm path length (Hellma, Forest Hills, NY). Spectra were collected at 198–260 nm wavelength (far

UV) at 25 °C. Raw data was processed by smoothening, as per the manufacturer's instructions. Three independent experiments were performed with each sample.

## Thioflavin T (ThT) binding assay

To measure ThT binding, PRL and GAL solutions were diluted in the same buffer into 200 µl such that final concentration of each sample was 10 µM. 4µl of 1 mM ThT prepared in 20 mM Tris-HCl buffer, pH 8.0 was added into each sample. ThT fluorescence was probed after the immediate addition of ThT. The fluorescence experiment was carried out in Shimadzu RF5301 PC (Japan), with excitation wavelength at 450 nm and emission wavelength from 460 to 500 nm. For measuring both excitation and emission, the slit width was kept at 5 nm. Three independent experiments were performed for each sample. The fluorescence value at 480 nm were plotted against incubation time, which produces the sigmoidal growth curve of amyloid formation. These curves were then used for lag time calculation.

The lag time ($t_{lag}$) was calculated as per the published protocol (*Willander et al., 2012*):

$$y = y_0 + (y_{max} - y_0)/(1 + e^{-k(t-t1/2)}) \tag{1}$$

here y is the ThT fluorescence at any particular time point, $y_{max}$ is the maximum ThT fluorescence observed and $y_0$ is the ThT fluorescence at $t_0$ (initial time) and $t_{lag}$ was defined by as

$$t_{lag} = t^{1/2} - 2/k \tag{2}$$

## Congo red (CR) binding

A 5 µl aliquot of protein/peptide sample was added into 80 µl of 5 mM potassium phosphate buffer containing 10% ethanol. A total of 100 µM CR solutions were prepared in 5 mM phosphate (containing 10% ethanol) and 15 µl of the solution was added to the sample. After incubating for 15 min in dark, absorption spectra were taken from 300 to 700 nm (JASCO V-650 spectrophotometer). For control, CR solution without protein was also measured. Three independent experiments were performed for each sample.

## CR birefringence study

Protein fibrils were obtained by ultracentrifuging the fibril solution at 95,000 rpm for 1 hour followed by washing with Milli-Q water. The fibrils were mixed in 100 µl of alkaline sodium chloride solution for 20 min with vortexing, to ensure uniform mixing of all fibrils in solution. The mixture was further centrifuged and pellet fractions were stained with alkaline CR solution for 20 min with vortexing. After that, mixtures were again centrifuged at 95,000 rpm for 1 hr, and pellets were washed two times by 500 µl of 20% ethanol. The pellets were then resuspended in PBS and spotted onto glass slides and subjected to air-drying at room temperature. The slides were observed using a microscope (Olympus SZ61 stereo zoom) attached with two polarizers and a camera.

## Immunoelectron microscopy

A total of 10 µl of PRL-GAL or PRL-CSA fibril was spotted onto the TEM grid. Ten µl of rabbit anti-PRL antibody (1:10) and/or mouse anti-GAL antibody (1:10) in PBS was added to the fibrils and was incubated for 1 hr. The excess antibody was removed using filter paper. The grid was subsequently washed thrice with autoclaved Milli-Q water. Anti-mouse secondary antibody conjugated with 10 nm gold particles (1:200) and/or anti-rabbit secondary antibody conjugated with 5 nm gold particles (1:200) was added to the grid and incubated for 30 min. The grid was then washed thrice with Milli-Q water, followed by staining with 1% uranyl formate for 5 min and it was imaged using TEM (CM 200, Netherland), and analyzed using KEEN view software.

## Transmission electron microscopy (TEM)

The protein/peptide sample was diluted in Milli-Q water to ~60 µM. Then, the samples were spotted on a carbon-coated, glow-discharged Formvar grid (Electron Microscopy Sciences, Fort Washington, PA) and were kept for incubation for 5 min. The grids were further washed with Milli-Q water and were stained with a 1% (w/v) uranyl formate solution. TEM imaging was done using FEITecnai G$^2$ 12 electron microscope at either 120 kV or 200 kV with nominal magnifications in the range of 26,000–60,000.

Images were collected by using the SIS Megaview III imaging system. Independent experiments were carried out thrice for each sample.

## X-ray fibril diffraction

PRL/GAL fibrils were isolated by ultracentrifugation as mentioned earlier and were loaded into a clean 0.7 mm capillary. The samples in capillary were dried overnight under vacuum. The whole capillary with dried protein was placed in the path of X-ray beam. The dried film of protein was placed in an X-ray beam at 200 K for 120 s exposure. The resulting images were collected using a Rigaku R-Axis IV ++ detector (Rigaku, Japan) kept on a rotating anode. The distance between the sample to the detector was 200 mm and the image files were analyzed and processed using Adxv software.

## FTIR spectroscopy

For FTIR spectroscopy, isolated fibrils or monomers were spotted onto a thin KBr pellet and were subjected to dry under an IR lamp. Then the spectrum was collected using a Bruker VERTEX 80 spectrometer attached with a DTGS detector at the frequency range of 1800–1500 cm$^{-1}$, corresponding to amide I stretching frequency, with a resolution limit of 4 cm$^{-1}$. The recorded spectrum was deconvoluted at the frequency range 1700–1600 cm$^{-1}$, using Fourier Self Deconvolution (FSD) method and the deconvoluted spectrum was fitted using the Lorentzian curve fitting method using OPUS-65 software (Bruker, Germany) according to the manufacturer's instructions. Independent sets were performed thrice for each sample.

## Seeding and cross-seeding by different fibril-seeds

Amyloid formation by PRL/GAL was confirmed by ThT binding and TEM imaging. After that, various fibrils were collected separately via ultra-centrifugation, and each fibril sample was suspended in 20 mM phosphate buffer, pH 6.0, 100 mM NaCl, 0.01% sodium azide. These fibrils were then subjected to sonication (03 s 'on' and 01 s 'off' at 20% amplitude) for 10 min to obtain preformed fibril seeds, which were mixed (1%, 2%, and 5% (v/v)) in the respective monomeric protein for homo seeding or to the other protein for cross seeding. The time-dependent aggregation was probed by ThT binding and CD spectroscopy. The ThT fluorescence data for GAL aggregation by PRL seed was fitted using the web-based software interface Amylofit (*Meisl et al., 2016*). For each seed concentration, triplicate sets of data (from the initial time point to till the plateau of ThT fluorescence), was normalized using Amylofit. Fitting was done using Nucleation Elongation and Secondary Nucleation Dominated models, following guidelines stated by Meisl et al. (*Meisl et al., 2016*).

## Surface plasmon resonance (SPR) spectroscopy analysis of PRL-GAL interaction

GAL was immobilized on to Biacore CM5 sensor chip (GE Healthcare) via amine coupling. To do that, 300 µg/ml of GAL solution was made in 50 mM sodium acetate buffer, pH 5, and was injected at a rate of 10 µl/min for 720 s to achieve a response unit (RU) of 1252 RU. PRL protein was dissolved in 20 mM phosphate buffer containing 100 mM NaCl, pH 6.0 to obtain 1 mM stock solution and was used for preparing different dilutions (7.8, 15.6, 31.2, 62.4, 125, 250, and 500 µM). These solutions were then injected over the immobilized GAL at a flow rate of 45 µl/min for 60 s. The dissociation was initiated at a flow rate of 30 µl/min, and the signal was recorded for 300 s. The chip was further regenerated using a 10 mM NaOH solution. ACTH and GH protein were used to examine the binding with PRL and GAL, respectively using a CM5 chip. For immobilization, 500 µg/ml of ACTH solution was made in 50 mM sodium acetate buffer (pH 4.5). For PRL immobilization, 2 mg/ml of PRL protein in 50 mM sodium acetate buffer (pH 4) was used. A similar range of concentrations was used for ACTH and GH as mentioned before. All the binding experiments were performed at 37 °C. The data were subjected to reference subtraction to compensate for bulk refractive index differences. Further, the buffer system for all the proteins was constant to avoid bulk/viscosity changes. The kinetics data for all protein interactions were fitted using a 1:1 state kinetic model in the Biacore T200 Software. The $k_{on}$, $k_{off}$, and $K_D$ values were determined for better understanding of the protein-protein interactions. The fitted curves were plotted in the GraphPad Prism Software v8.4.2.

## In silico study of PRL and GAL interaction and co-aggregation

Protein-protein docking and all-atom molecular dynamics (MD) simulations were used to probe the co-aggregation propensities (if any) and resulting secondary structural transitions in PRL and GAL

molecules. For this, the initial structure of PRL was obtained from PDB ID: 1RW5 (*Teilum et al., 2005*). Since there was no entry for GAL structure in PDB, the GAL structure was built from its primary amino acid sequence and energy minimized. Subsequently, the minimized structure was equilibrated and simulated for 300ns. Initially, GAL was docked at two amyloidogenic regions of PRL (residues 18–28) (set-1) and (residues 80–88) (set-2), which were predicted by TANGO (*Fernandez-Escamilla et al., 2004*). The docking was performed by the protein-protein docking program, HADDOCK (*van Zundert et al., 2016*). The lowest energy complexes were extracted from each set, denoted as complex 1 from set-1 and complex 2 from set-2. These two complexes were used as starting structures for the MD simulations. The first sets of simulations were performed using the AMBER16 package with the Amberff99SB force field. The LEAP module of AMBER16 was used to add the hydrogen for the heavy atoms. The complexes were then energy minimized for 2000 steps using the steepest descent and conjugate gradient algorithms. Subsequently, the structures were hydrated in a cubic periodic box extending 9 Å outside the protein-protein complex on all sides with explicit water molecules. The three-site TIP3P model was chosen to describe the water molecules. The charge of each system was neutralized by placing Na$^+$ ions randomly in the simulation boxes. The systems were again minimized to prevent any random contacts formed due to the solvation. All the systems were then equilibrated for 500 ps in NVT ensemble at 300 K followed by one ns in NPT ensemble at 1 atm of pressure. After the density and potential energy of the systems had converged, each complex was subjected to 300 ns of the production run. To further validate our simulations, we performed a new set of simulations using different force fields. Each complex was subjected to a 250 ns simulation using the GROMOS 53a6 force field. The new sets of simulations were performed using the GROMACS package following the above-mentioned protocol. VMD tool was used for visual analysis of the trajectories.

To further strengthen our results from MD simulations, we have performed computational mutagenesis studies on complex 2 from the Amberff99SB force field. A total of four cases were examined – case 1: four GAL residues that were involved in the β-sheet formation were mutated to alanine (W2A, T3A, L4A, N5A); case 2: four PRL residues that were involved in the β-sheet formation were mutated to alanine (I146A, Y147A, P148A, V149A). Furthermore, we simulated a single mutation in PRL (Y147P), since proline is known as β-sheet breaker (*Soto et al., 1998*) (case 3), and one deletion system by deleting the first two N-terminal residues of GAL (case 4). We did not simulate the PRL deletion system, as deletion at the middle of a structure would bring in conformational changes in itself, and correlating that with our interest in PRL-GAL β-sheet formation will be difficult.

MD simulations of all the systems were performed by following the protocol described in the previous section using the AMBER16 package with the Amberff99SB force field. While the mutated systems, systems 1–3 had to be simulated for ~400 ns to see the change, the deletion system showed the disappearance of the β-sheet rather quickly in 100–150 ns.

## Monomer release assay

Amyloid fibrils of PRL and GAL formed in the presence of glycosaminoglycans, and co-aggregated fibrils of PRL-GAL were harvested by ultracentrifugation at 90,000 rpm for 1 hr. The concentration of the soluble fraction (supernatant) was calculated using the absorbance at 280 nm (Jasco V-650) and was used to determine the concentration of the pelleted fibrils. A total of 100 µl re-dissolved pellet of 400 µM concentration was chosen to examine the monomer release study using the experimental setup, which has been reported previously (*Jacob et al., 2016*; *Maji et al., 2008*). 400 µM PRL and GAL solutions incubated for 15 days were used as a monomer control. Briefly, for PRL monomer release, the pellet solutions (PRL, PRL-CSA, and PRL-GAL) were transferred into a modified PCR tube with a pierced hole in its cap, which was attached and sealed with a 50 kDa molecular weight cutoff membrane (Pierce, USA). This whole setup was then placed inside a 1 ml cryotube (Nunc, Denmark) containing 500 µl of 10 mM Tris-HCl buffer (pH 7.4), 0.01% sodium azide. Meanwhile, to investigate the release of GAL monomers from the fibril of GAL-Hep, pellet solution was placed in a 10 kDa cutoff Slide-A-Lyzer mini dialysis unit system (Pierce, USA), which was positioned onto a 1 ml cryo-tube (Nunc, Denmark) containing 500 µl of Tris-HCl buffer (pH 7.4), 0.01% sodium azide. The tubes were then kept at 4 °C to prevent evaporation of solutions. To calculate the concentration of the protein outside of the membrane, an aliquot of 100 µl of the solution was taken from the buffer outside of the dialysis membrane at different times, and absorbance was measured at 280 nm. For measuring the monomer release of PRL and GAL from PRL-GAL co-aggregates, 100 µl releasing medium was taken

out, and the same volume of fresh buffer was added back as volume correction (subsequent concentration correction has been done in the data). The separated 100 µl releasing medium was passed through a 10 kDa centrifugal filter unit (Amicon Ultra, Millipore). After which, the GAL monomer was obtained as filtrate, and the PRL was recovered from the retentate by washing the reversed filter unit with 100 µl of 10 mM Tris, pH 7.4 as per the manufacturer's protocol. In an identical experimental setup, the absorbance was measured from the buffer both inside and outside of the membrane. This control is kept to check whether degradation of the materials of the dialysis membrane is interfering with the assay. Each time, the released solution was returned after the spectra recording. Three independent experiments were performed for each sample.

## Cell proliferation assay using Nb2 cells

Nb2 cell line was grown in plastic culture flasks in RPMI medium (HiMedia, India) supplemented with 10% heat-inactivated fetal bovine serum (Gibco, USA), 10% horse serum (HiMedia, India), and 1 X antibiotic solution and incubated at 37 °C in a humidified incubator containing 5% $CO_2$ in the air. The Nb2 cell line was tested as mycoplasma negative. Since there is no publicly available database for rat-derived cell lines to be used as a reference, the STR profiling was not employed. However, the authentication of the Nb2 cell line was done by monitoring the morphology and the characteristic property of the cell line (comprising PRL receptors) to proliferate in presence of PRL as evident from our cell proliferation study. For proliferation assay, cells ( ~ $10^5$/well) in a 96-well plate were seeded in RPMI medium in the presence of 1% fetal bovine serum and 10% horse serum and incubated for 24 hr to synchronize the cells at G0/G1 phase. After 24 hr, cells were treated with PRL monomer, monomer released from PRL-CSA and PRL-GAL at a dose range from 0.5 to 4 µM. The unrelated protein ovalbumin was used as the negative control. After incubation, cell proliferation was measured by MTT assay. To do so,10 µl of MTT solution (5 mg/ml in PBS) was added to the cells and incubated for 4 h. Subsequently, 100 µl of SDS-DMF solution (50% DMF and 20% SDS, pH 4.75) was added for overnight incubation. The absorption value of the product was measured at 560 nm and 690 nm as a background absorbance using a Spectramax M2e microplate reader (Molecular Devices, USA). The fold increases in cell proliferation compared to untreated cells were plotted against the concentration of the sample administered.

## Immunofluorescence

Adult, Sprague-Dawley rats (200–250 g) (both male and female) taken for this study were maintained under the standard environmental conditions (12 h: 12 h, light: darkness cycle, chow, and water ad libitum). The Institutional Animal Ethical Committee (IAEC) at NISER, Bhubaneswar (ethical approval protocol number: NISER/SBS/AH-210 and NISER/SBS/AH-212), under the Committee for the Purpose of Control and Supervision of Experiments for Animals (CPCSEA), New Delhi, India, approved the experimental protocol. First, the animals were anesthetized with a mixture of ketamine and xylazine and perfused transcardially with 50 ml of 10 mM phosphate buffer saline (PBS, pH 7.4), followed by 100 ml 4% paraformaldehyde (PFA) in 100 mM phosphate buffer (pH 7.4). The pituitary glands were dissected out and post-fixed in 4% PFA overnight at 4°C followed by immersion in 25% sucrose solution in PBS for 24 hr at 4°C. The pituitary glands were rapidly frozen in powdered dry ice, sectioned on the cryostat (Leica, CM3050 S), and sections were mounted on poly-L-lysine (Sigma) coated glass slides. The sections were processed for double immunofluorescence. The sections were rinsed twice in PBS/TBS followed by 0.5% Triton X-100 for 20 min. The sections were incubated in blocking solution for 30 min. The double immunofluorescence was performed in a humidified chamber by incubating the sections in a mixture of anti-PRL (Guinea pig polyclonal, from A. F. Parlow, National Hormone, and Pituitary Program, Harbor-ULCA Medical Center, Torrance, CA, 1:1500) and mouse polyclonal anti-GAL antibody (Abcam, dilution 1:1500) overnight at 4 °C. Further, co-staining of pituitary tissue amyloids of GAL or PRL was performed using amyloid-specific (OC) antibody (rabbit polyclonal, Abcam, 1:500) and with respective hormone antibodies. All the primary antibody-stained tissue slices were incubated overnight at 4 °C in a humidified chamber. The sections were rinsed in TBST and further incubated with the secondary antibody of goat anti-mouse FITC (1:500) or goat anti-rabbit FITC (1:500) or goat anti-mouse Alexa Fluor-555 (1:500) (Thermo Scientific, USA) or goat anti-Guinea pig Alexa Fluor 555 (1:500) for 2 h at room temperature in a humidified chamber. The sections were washed with TBST and mounted with a mounting medium. The sections were analyzed using a confocal microscope (Olympus

IX81 combined with FV500) (Shinjuku, Tokyo, Japan) and images were recorded using a multi-channel image acquisition tool of Fluovision software (Zeiss, Oberkochen, Germany). Similarly, the GH and ACTH double immunofluorescence was performed using rabbit polyclonal anti-ACTH (kind gift of Dr. A. F. Parlow, NHPP, dilution 1:1000) and goat polyclonal anti-GH (R&D Systems, dilution 1:1000) overnight at 4°C. The sections were washed and further incubated in a mixture of secondary antibodies [Alexa Fluor 594 anti-rabbit IgG or Alexa Fluor 488 anti-goat IgG (1:500, Thermo Fisher Scientific, USA)] for 2 h at room temperature. The sections were rinsed and sealed with mounting medium, and the association between GH and ACTH analyzed under an AxioImager M2 fluorescence microscope attached with AxioCamMRm digital microscope camera (Carl Zeiss, Göttingen, Germany). The microscopic images of pituitary sections showing GH and ACTH were captured using the same microscope and camera. For Thioflavin S (ThioS) staining, GAL and PRL staining was done using goat anti-mouse Alexa Fluor-555-conjugated secondary antibody or goat anti-Guinea pig Alexa Fluor 555 (1:500 dilution) respectively. The sections were then stained with 0.6% ThioS (Sigma-Aldrich) for 5 min in dark. The sections were washed with 50% ethanol for 2 min followed by TBST washing for 3 min. The slides were then mounted in 90% glycerol and 10% PBS containing 1% DABCO (1, 4-diazabicyclo-[2.2.2] octane, Sigma-Aldrich). The images and the sections were analyzed by multi-channel image acquisition tool of Fluovision software (Zeiss, Oberkochen, Germany) and Olympus FV-500 IX 81 confocal microscope (Shinjuku, Tokyo, Japan), respectively.

## Acknowledgements

The authors wish to acknowledge Prof. PS Dannies and Prof. ME Hodsdon, Yale School of Medicine, USA for the plasmid of PRL. We are also grateful to Prof. Amulya K Panda, NII, India for the kind gift of Nb2 cell line. We would like to acknowledge Dr. Srivastav Ranganathan and Dr. Prem Prakash for the PRL schematic drawing and Congo red birefringence study, respectively. We are also thankful to CRNTS and IRCC, IIT Bombay for FTIR, electron microscopy, confocal microscopy, protein crystallography facility, and SPR facility. Authors wish to acknowledge DBT (BT/PR9797/NNT/28/774/2014) Government of India, Wadhwani research centre for Bioengineering (WRCB), and DBT/Welcome Trust India Alliance Fellowship [RD/0119-DBTFL49-001] awarded to Shinjinee Sengupta for financial support.

## Additional information

### Funding

| Funder | Grant reference number | Author |
| --- | --- | --- |
| Department of Biotechnology , Ministry of Science and Technology | BT/PR9797/NNT/28/774/2014 | Samir K Maji |
| Department of Biotechnology , Ministry of Science and Technology | BT/HRD/35/01/03/2020 | Samir K Maji |
| Department of Science and Technology, Ministry of Science and Technology | CRG/2019/001133 | Samir K Maji |
| India Alliance | RD/0119-DBTFL49-001 | Shinjinee Sengupta |

The funders had no role in study design, data collection and interpretation, or the decision to submit the work for publication.

### Author contributions

Debdeep Chatterjee, Conceptualization, Data curation, Formal analysis, Methodology, Validation, Visualization, Writing - original draft, Writing - review and editing; Reeba S Jacob, Data curation, Formal analysis, Methodology, Validation, Visualization; Soumik Ray, Formal analysis, Validation, Visualization, Writing - review and editing; Ambuja Navalkar, Namrata Singh, Shinjinee Sengupta, Data curation, Formal analysis, Methodology; Laxmikant Gadhe, Pradeep Kadu, Data curation, Methodology;

Debalina Datta, Formal analysis, Methodology; Ajoy Paul, Surabhi Mehra, Chinmai Pindi, Santosh Kumar, Data curation; Sakunthala Arunima, Investigation, Methodology, Writing - review and editing; Praful Singru, Sanjib Senapati, Validation; Samir K Maji, Conceptualization, Funding acquisition, Supervision, Writing - review and editing

### Author ORCIDs
Debdeep Chatterjee http://orcid.org/0000-0001-8153-8651
Surabhi Mehra http://orcid.org/0000-0003-1777-673X
Samir K Maji http://orcid.org/0000-0002-9110-1565

### Ethics
Adult, female, Sprague-Dawley rats taken for this study were maintained under the standard environmental conditions and Institutional Animal Ethical Committee (IAEC) at NISER, Bhubaneswar, India approved the experimental protocols. (Protocol Numbers: NISER/SBS/AH-210 and NISER/SBS/AH-212).

### Decision letter and Author response
Decision letter https://doi.org/10.7554/eLife.73835.sa1
Author response https://doi.org/10.7554/eLife.73835.sa2

## Additional files

### Supplementary files
• Transparent reporting form

### Data availability
All data generated or analysed during this study are included in the manuscript and supporting file. Source Data files have been provided for main Figures 1-5 and Supplementary figures and tables.

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
