## [Editor Report]

This study unravels the formation of prolactin/galanin functional amyloids and their storage in secretory granules of the anterior pituitary gland. Our understanding of the regulation of hormonal release from the pituitary gland is improved with this report. It will be of interest to the field of endocrinology, neurobiology and cancer.

---

## [Decision Letter]

**Decision letter after peer review:**

Thank you for submitting your article "Co-aggregation and secondary nucleation in the life cycle of human prolactin/galanin functional amyloids" for consideration by *eLife*. Your article has been reviewed by 3 peer reviewers, one of whom is a member of our Board of Reviewing Editors, and the evaluation has been overseen by Mone Zaidi as the Senior Editor. The following individual involved in review of your submission has agreed to reveal their identity: Daniel Erik Otzen (Reviewer #3).

Essential revisions:

1) Consider testing for colocalization of ACTH and GH

2) Consider analyzing the data with chemical kinetics

3) Consider discussing the physiologic implications of the findings

*Reviewer #1 (Recommendations for the authors):*

What are the physiologic implications of the findings? If co-aggregation of PRL-GAL is impaired and secretion is slowed, would this inhibit normal prolactin-mediated effects (lactation) or normal galanin-mediation effects (regulation of energy metabolism)? Could this have therapeutic implications for the treatment of prolactinoma? What are the functional implications of the finding that PRL amyloid fibers can cross-seed GAL monomers, but that GAL fibrils can't cross-seed PRL monomers? It would be useful to see if PRL-GAL facilitates synergistic aggregation to amyloid fibers in male rats in addition to female rats.

*Reviewer #2 (Recommendations for the authors):*

The manuscript is well written and easy to understand. All the results are explained clearly and the conclusions are drawn based on the provided results only. There are no major changes that need to be made. However, I would like to make a few suggestions in order to improve the manuscript.

– Although the term GAG is very familiar, incorporating the expanded form of GAG (Glycosaminoglycans) in the manuscript makes the readers more comfortable.

– In Line 180: "We hypothesized that there could be 4 possibilities (case 1-4) when PRL and GAL are co-stored as amyloids".

The authors had mentioned about 4 cases and they have shown it as 5 cases in line 186 and Figure 2a.

– In line 669: "After harvesting the cells at 8000 rpm for 20 minutes, it was dissolved in 20 mM Tris HCl, pH 8.0".

The fact that the cells were dissolved confuses the general reader.

– Authors should be appreciated for efforts to understand the Prolactin Galanon interactions at the atomic level by molecular dimensions simulations. Although it is not required as per the claims that authors have made, it might be better if they show a loss of interactions when the proposed site of interactions is mutated or deleted.

*Reviewer #3 (Recommendations for the authors):*

I only have two major comments for the authors to reflect on.

1. The authors identify colocalization of PRL and GAL in Figure 1d. This is fine but potentially rather limited. What other hormones are stored in this tissue and could they conceivably be colocalized with these deposits? I.e. are we missing the bigger picture of higher-order colocalization? Did the authors test for other hormones as well or did they make an educated guess from the start? At least the prospect should be addressed. The authors subsequently examine ACTH and GH but it is not clear whether these hormones could be present in e.g. Figure 1d. Have the authors tested for colocalization of these hormones?

2. The authors present good arguments for secondary nucleation rather than elongation/fragmentation based on data in Figure 3. However, additional information could possibly be provided by analyzing the data with chemical kinetics using web-server programs such as Amylofit or designing experiments that might in principle distinguish between these options. I would like the authors to explore this further.

---

## [Author Response]

Essential revisions:1) Consider testing for colocalization of ACTH and GH

Thank you for suggesting this revision. We have tested the colocalization of ACTH and GH and the results are now incorporated in the revised manuscript.

2) Consider analyzing the data with chemical kinetics

We have analyzed the chemical kinetics data with Amylofit and the results are incorporated in the revised manuscript*.*

3) Consider discussing the physiologic implications of the findings

We have now discussed the physiologic implications of the findings, which are incorporated in the revised manuscript.

Reviewer #1 (Recommendations for the authors):What are the physiologic implications of the findings?

We thank you for allowing us to describe the biological implications of this study. It is known that intermolecular interaction (such as hormone-hormone, hormone-receptor), cross talk, and feedback (both positive and negative) are the important mechanisms, which are associated with hormone regulation in mammals. Further, protein/peptide hormone synthesis and their storage/secretion are controlled by various secretagogues/cellular factors. These mechanisms, often in combination, maintain normal physiological functions of hormones. In the context of the present study, peptide hormone galanin (GAL), which is known to be highly expressed in the hypothalamus performs various tropic activities including regulating the secretion of other neuropeptide hormones and neuronal differentiation^1^. Interestingly, GAL is overexpressed in the anterior pituitary during lactation^2^, which is known to regulate prolactin (PRL) expression, storage, and release^3,4^. Further, storing a high amount of PRL in the anterior pituitary during lactation is important for the development of lactotrophs^4^. In this line, GAL deficiency or functional mutation has been shown to inhibit PRL release as well^4^—suggesting its crucial role in PRL regulation in mammals. Naturally, efficient storage and release of GAL are necessary for PRL homeostasis.

However, basal expression of GAL in the endocrine cells of the anterior pituitary is low^3^. Further, not all PRL producing cells can synthesize GAL and its half-life (turnover cycle) is low (~3-4 min in circulation^5^) possibly due to its short length and unstructured nature compared to PRL (~41 min)^6,7^. Interestingly, irrespective of GAL secretion by a minority of lactotrophs (~ 9% of cells)^3^, it is however required to maintain its basal level for regulating the secretion of other neuropeptides such as PRL and VIP^3^. Moreover, lactotrophs that do not produce GAL are highly sensitive to this hormone but not to other secretagogues like VIP peptide^3^, suggesting the essential requirement of GAL for several pituitary functions. Therefore, efficient storage of GAL is essential for PRL storage and regulated release in absolute quantities as required during pregnancy and lactation. Together, these facts indicate that the physiological life cycle and functionality of PRL and GAL are complementary.

In this line, our data clearly showed GAL and PRL are interdependent on their storage inside the same secretory granules in high amounts in the amyloid-like state, ensuring the controlled release of these hormones. When GAL monomers release from the co-aggregates (PRL-GAL), it ensures further release of PRL and other tropic factors both in an autocrine and paracrine manner^8,9^. The co-aggregation and unidirectional cross-seeding mechanism not only supports PRL storage and release but also helps to maintain a very low titre of unstable GAL to efficiently store in secretory granules for future use. Thus, a substantial decrease in physiological PRL levels would inevitably affect the storage of GAL and its subsequent release as they are in a positive feedback loop. On the other hand, over-production of GAL may result in hypersecretion of prolactin—leading to prolactinoma. Therefore, tight regulation is required for GAL synthesis and secretion. Taken together, our findings probe that co-aggregation and co-storage would allow much higher efficiency for hormone release and function than their individual storage.

We have now incorporated the physiological significance of our findings in the Discussion section of the revised manuscript.

If co-aggregation of PRL-GAL is impaired and secretion is slowed, would this inhibit normal prolactin-mediated effects (lactation) or normal galanin-mediation effects (regulation of energy metabolism)?

Thank you. This is an interesting point and we appreciate this query from you. Our study probes that PRL-GAL co-aggregation is one of the key mechanisms for PRL and GAL storage inside the secretory granules and subsequent release of PRL/GAL functional hormones. In this line, earlier studies showed that the generation of mice carrying a loss of function mutation in the endogenous *GAL* gene has shown a drastic decrease in PRL level compared to the wild-type controls^4^. This was accompanied by reduced PRL secretion, failure in mammary gland maturation, and loss of lactotroph proliferation upon estrogen treatment in the mutant mice after 7 days of postpartum compared to wild-type controls^4^. On the other hand, estrogen treatment drastically enhanced the GAL mRNA by several orders of magnitude^2,4^ (also responsible for lactotroph proliferation and subsequent PRL over-expression), suggesting the common secretagogue for PRL and GAL. However, functional implications of GAL by loss of PRL synthesis/function are yet to be established.

It is important to note that PRL and GAL are co-stored in the lactotrophs of the female rat anterior pituitary^10,11^. Indeed, there are other pathways for homotypic PRL/GAL aggregation and subsequent storage in presence of glycosaminoglycans like CSA or *Hep*, respectively. However, homotypic aggregation might be insufficient to maintain the entire PRL/GAL protein homeostasis inside the living body, which is normally achieved by PRL-GAL co-aggregation. With the current knowledge, it is evident that impairment of PRL-GAL co-aggregation would very likely be detrimental in achieving the PRL mediated lactation and other tropic functions of GAL.

We have now discussed these points in the revised Discussion section of the manuscript.

Could this have therapeutic implications for the treatment of prolactinoma?

Our study indeed has potential therapeutic implications for the treatment of prolactinoma, which is caused by hypersecretion of PRL. Studies have shown that transgenic mice containing mouse GAL gene fused to rat PRL promoter causes hyperplasia and hyperprolactinemia (symptoms of prolactinoma) with a dramatic increase in GAL^12^. It has been well established that PRL secretion is directly governed by GAL secretion because over-expression of GAL, in most cases, is accompanied by over-expression of PRL^3,13,14^.

It is important to note that PRL and GAL cross-talk via a positive feedback loop for both storage and release. GAL, once released, causes further release of PRL and other tropic factors^8^ via both autocrine and paracrine manner^9^. Therefore, a decrease in GAL level would also decrease the storage and release of PRL. This is also evident from our results indicating the co-aggregation and unidirectional cross-seeding data. Exploiting this phenomenon, one could argue that by controlling (reducing) excess GAL levels, it is possible to prevent PRL hypersecretion (as seen for prolactinoma). This could, in principle, be achieved by targeting excess GAL in the bloodstream using specific antibodies targeted for proteolysis or by engineering small molecules/metabolites capable of inhibiting GAL secretion as previously demonstrated for various hormones^15-18^. Further, instead of targeting GAL, other metabolites such as Tryptophan amino acid can also be used that target PRL directly to control its release^19^.

One more interesting aspect of functional amyloids is that they must exhibit sustained release of functional monomers at the target site. It will be interesting to try to generate GAL analogs that can form stable co-amyloids with PRL, which would be incapable of releasing PRL monomers—thereby controlling the excess secretion of PRL. However, translating the current knowledge into therapeutics has challenges—primarily because, although PRL-GAL co-aggregation and unidirectional cross-seeding is very efficiently designed by nature, both hormones can also form functional amyloids independently in the presence of helper molecules such as GAGs.

We have now discussed this in detail in the revised Discussion section of our manuscript.

What are the functional implications of the finding that PRL amyloid fibers can cross-seed GAL monomers, but that GAL fibrils can't cross-seed PRL monomers?

Thank you for raising this interesting aspect. As we mentioned in response to question-1 the activities of GAL in the pituitary depend on its efficient storage and sufficient release. However, basal expression of GAL in the endocrine cells of the anterior pituitary is low^3,4^ (unless overproduce by circulating steroid hormone, estrogen)^3^. Despite GAL being the most important factor for PRL release, not all PRL producing cells in the anterior pituitary produce GAL^3^. In addition, the lifetime of GAL is relatively low (~ 3-4 minutes)^5^ in the bloodstream due to its short length and unstructured nature compared to PRL (~ 41 minutes)^6,7^. Therefore, efficient storage and release of GAL is physiologically highly relevant. Cross-seeding by PRL amyloids increases the chances of GAL storage and subsequent release. On the other hand, PRL storage needs tight regulation due to the risk of developing prolactinoma upon over-secretion.

From a thermodynamic point of view, templating mechanism requires the client monomers to change their fold—which requires a substantial amount of energy. It is very likely that for a folded protein like PRL, it is thermodynamically less favorable to undergo a conformational transition upon templating by GAL seeds. On the other hand, GAL is an unstructured neuropeptide and can easily be templated by PRL seeds. This is probably an evolutionary optimized design by nature so that tight regulation of multiple hormones storage and functions can be regulated.

We have discussed these points in the revised Discussion section of our manuscript.

It would be useful to see if PRL-GAL facilitates synergistic aggregation to amyloid fibers in male rats in addition to female rats.

We thank you for suggesting this important experiment. Notably, earlier studies on rat models have already shown that PRL and GAL co-localize exclusively in the lactotrophs (PRL secreting cells) of the female (and not in the male) rat anterior pituitary^10,11^. Based on this, we had limited our study to examine PRL/GAL co-aggregation only in the female rat anterior pituitary in our initial submission. However, after your suggestions, we have now performed the double immunofluorescence study of PRL/GAL colocalization in anterior pituitary of male rats. Consistent with previous studies, we also observed no significant colocalization of PRL/GAL (incorporated in the revised manuscript).

Reviewer #2 (Recommendations for the authors):The manuscript is well written and easy to understand. All the results are explained clearly and the conclusions are drawn based on the provided results only. There are no major changes that need to be made. However, I would like to make a few suggestions in order to improve the manuscript.– Although the term GAG is very familiar, incorporating the expanded form of GAG (Glycosaminoglycans) in the manuscript makes the readers more comfortable.

We have now replaced the term ‘GAG’ with ‘glycosaminoglycans’ throughout the revised manuscript.

– In Line 180: "We hypothesized that there could be 4 possibilities (case 1-4) when PRL and GAL are co-stored as amyloids".The authors had mentioned about 4 cases and they have shown it as 5 cases in line 186 and Figure 2a.

We apologize for this confusion. We want to emphasize that Case 4 (PRL monomers adhere to the GAL fibrils) and Case 5 (GAL monomers adhere to the PRL fibrils) are in essence vice-versa and fall under the same category. As per your suggestions, we have now replaced the case 4 and 5 terms in the manuscript as case 4a or 4b, respectively.

– In line 669: "After harvesting the cells at 8000 rpm for 20 minutes, it was dissolved in 20 mM Tris HCl, pH 8.0".The fact that the cells were dissolved confuses the general reader.

The cells were pelleted down by centrifugation at 8000 rpm for 20 minutes and the pellet was subsequently dispersed in 20 mM Tris HCl, pH 8.0. We have now rephrased the sentence as:

"After harvesting the cells at 8000 rpm for 20 minutes, the pellet was dispersed in 20 mM Tris HCl, pH 8.0".

– Authors should be appreciated for efforts to understand the Prolactin Galanon interactions at the atomic level by molecular dimensions simulations. Although it is not required as per the claims that authors have made, it might be better if they show a loss of interactions when the proposed site of interactions is mutated or deleted.

We thank you for raising a very interesting point. To address your queries, we have now done the MD simulation with various mutations/deletions to either of these hormones**.** We have chosen the complex 2 structure obtained from the Amber ff99SB force field to examine the effect of change in amino acids in hormones. We chose this structure as the formation of β-sheet is evident in complex 2 structures (GAL docked near the 80-88 residues of PRL) when the simulation is done using two different force fields (Amber ff99SB and GROMOS 53a6). We have mutated the residues of both PRL and GAL, which are involved in the formation of β-sheet to alanine, and also introduce a point mutation, to examine if there is a loss of interaction and the β-sheet formation. After altering these changes in complex 2 structure in Amber force field, we further simulated 400 ns simulation and examine the structural changes. We used the following alterations:

Case 1. GAL residues 2 to 5 (WTLN), which showed to form β-sheet after interaction with PRL are mutated to alanine as W2A, T3A, L4A, N5A.

Case 2. PRL residues 146 to 149 (IYPV), which showed to form β-sheet after interaction with GAL are mutated to alanine as I146A, Y147A, P148A, V149A.

Case 3. A point mutation is introduced at the 147^th^ position of PRL with proline (β-sheet forming residues are 146 to 149) which is Y147P. The proline was chosen as it is known as a β-sheet breaker^20^ and our goal was to examine if there is a loss of interaction due to substitution of tyrosine by proline, which would disrupt the β-sheet formation.

Case 4. N-terminal first two (GW) amino acid residues were deleted. These N-terminal amino acids were deleted as N-terminal 2-5 residues of GAL are involved in the PRL-GAL β-sheet formation in the complex 2.

We did not simulate the PRL deletion system, as deletion at the middle of the PRL helical structure (PRL residues 146 to 149) would bring in conformational changes in the PRL secondary structure itself, and correlating that with our interest of PRL-GAL β-sheet formation will be difficult.

Interestingly, in all the mutated systems, we observed loss of interactions (H-bonds) at the PRL-GAL interface, which eventually led to the loss of β-sheet structures (incorporated in the revised manuscript). It is worth mentioning here that a single mutation in PRL (Y147P; Case 3) was sufficient to disrupt the secondary structure formation, suggesting its key role in inducing the formation of β-sheet during co-aggregation of PRL and GAL (incorporated in the revised manuscript). Furthermore, deletion of N-terminal residues in GAL (Case 4) also resulted in the loss of β-sheet formation (incorporated in the revised manuscript). Thus, our MD simulation results convincingly show that the mutation or deletion at the PRL-GAL interface could result in the loss of β-sheet formation, which eventually affects the PRL-GAL co-aggregation. Now we have included this data in the revised manuscript with relevant discussions.

Reviewer #3 (Recommendations for the authors):I only have two major comments for the authors to reflect on.1. The authors identify colocalization of PRL and GAL in Figure 1d. This is fine but potentially rather limited. What other hormones are stored in this tissue and could they conceivably be colocalized with these deposits? I.e. are we missing the bigger picture of higher-order colocalization? Did the authors test for other hormones as well or did they make an educated guess from the start? At least the prospect should be addressed. The authors subsequently examine ACTH and GH but it is not clear whether these hormones could be present in e.g. Figure 1d. Have the authors tested for colocalization of these hormones?

The query is very important in understanding the implication of our work and we thank you for asking this important question. The hormones produced in the anterior pituitary are Growth hormone (GH), Prolactin (PRL), Adrenocorticotropic hormone (ACTH), Thyroid-stimulating hormone (TSH), Follicle-stimulating hormone (FSH), and Luteinizing hormone (LH) and there can be at least 6x5=30 combinations by which two individual hormones can co-localize. However, it is also possible that more than 2 hormones can co-localize, which increases the possible combinations to a very large, unfeasible number to experimentally explore.

However, there are ample evidence that GAL over-expression enhances PRL release and PRL/GAL is released from a common secretagogue^4,13,14^. In this line, you are right that we indeed made an ‘educated guess’ by picking up PRL and GAL for studying their co-aggregation. This was primarily because PRL and GAL were found in the same secretory granules of the female anterior pituitary^10,11^.

However, after your suggestion, we have now tested for the co-localization for GH and ACTH in the female rat anterior pituitary using immunofluorescence. We found that GH and ACTH do not co-localize with each other (incorporated in the revised manuscript). This indeed increases the relevance of our findings and we thank you for this suggestion. We have now included this discussion and results in the revised manuscript.

2. The authors present good arguments for secondary nucleation rather than elongation/fragmentation based on data in Figure 3. However, additional information could possibly be provided by analyzing the data with chemical kinetics using web-server programs such as Amylofit or designing experiments that might in principle distinguish between these options. I would like the authors to explore this further.

We thank you for this important suggestion. As per your suggestion, we have now performed the global fitting analysis of the PRL-GAL cross-seeding kinetics data at different PRL seed concentrations (1%, 2%, and 5%) using Amylofit^21^ (version 2.0), which has been extensively used to identify the underlying mechanism behind the kinetics of amyloid aggregation^22-25^**.** We observed the data poorly fit with the nucleation-elongation model, which considers primary nucleation and elongation events. However, the kinetics data of GAL aggregation in presence of various PRL seeds (1%, 2%, and 5%) could be satisfactorily fit with the secondary nucleation model, which considers surface catalyzed secondary nucleation, along with classical elongation. The mean residual error (M.R.E.) in fitting in the elongation mechanism was observed to be higher compared to the fitting using the model with surface catalyzed secondary nucleation. The fitting data were in line with our hypothesis of surface-mediated amyloid assembly playing a dominant role in the aggregation mechanism of GAL in the presence of PRL seed (incorporated in the revised manuscript). We have now added this fitted data along with relevant discussions.

References

1. Lundstrom, L., Elmquist, A., Bartfai, T. and Langel, U. Galanin and its receptors in neurological disorders. *Neuromolecular medicine* 7, 157-180, doi:10.1385/NMM:7:1-2:157 (2005).

2. Kaplan, L. M. *et al.* Galanin is an estrogen-inducible, secretory product of the rat anterior pituitary. *Proc Natl Acad Sci U S A* 85, 7408-7412, doi:10.1073/pnas.85.19.7408 (1988).

3. Wynick, D., Hammond, P. J., Akinsanya, K. O. and Bloom, S. R. Galanin regulates basal and oestrogen-stimulated lactotroph function. *Nature* 364, 529-532, doi:10.1038/364529a0 (1993).

4. Wynick, D. *et al.* Galanin regulates prolactin release and lactotroph proliferation. *Proc Natl Acad Sci U S A* 95, 12671-12676, doi:10.1073/pnas.95.21.12671 (1998).

5. Hinghofer-Szalkay, H. G. *et al.* Circulatory galanin levels increase severalfold with intense orthostatic challenge in healthy humans. *Journal of applied physiology* 100, 844-849, doi:10.1152/japplphysiol.01039.2005 (2006).

6. Yoshida, Y. *et al.* [A kinetic study on serum prolactin concentration in the thyrotropin-releasing hormone test]. *Kaku igaku. The Japanese journal of nuclear medicine* 28, 585-590 (1991).

7. Yu, S., Alkharusi, A., Norstedt, G. and Graslund, T. An in vivo half-life extended prolactin receptor antagonist can prevent STAT5 phosphorylation. *PLoS One* 14, e0215831, doi:10.1371/journal.pone.0215831 (2019).

8. Vrontakis, M. E. Galanin: a biologically active peptide. *Current drug targets. CNS and neurological disorders* 1, 531-541, doi:10.2174/1568007023338914 (2002).

9. Cai, A., Bowers, R. C., Moore, J. P., Jr. and Hyde, J. F. Function of galanin in the anterior pituitary of estrogen-treated Fischer 344 rats: autocrine and paracrine regulation of prolactin secretion. *Endocrinology* 139, 2452-2458, doi:10.1210/endo.139.5.6025 (1998).

10. Steel, J. H. *et al.* Galanin and vasoactive intestinal polypeptide are colocalised with classical pituitary hormones and show plasticity of expression. *Histochemistry* 93, 183-189, doi:10.1007/BF00315973 (1989).

11. Hyde, J. F., Engle, M. G. and Maley, B. E. Colocalization of galanin and prolactin within secretory granules of anterior pituitary cells in estrogen-treated Fischer 344 rats. *Endocrinology* 129, 270-276, doi:10.1210/endo-129-1-270 (1991).

12. Cai, A., Hayes, J. D., Patel, N. and Hyde, J. F. Targeted overexpression of galanin in lactotrophs of transgenic mice induces hyperprolactinemia and pituitary hyperplasia. *Endocrinology* 140, 4955-4964, doi:10.1210/endo.140.11.7120 (1999).

13. Koshiyama, H. *et al.* Central galanin stimulates pituitary prolactin secretion in rats: possible involvement of hypothalamic vasoactive intestinal polypeptide. *Neuroscience letters* 75, 49-54, doi:10.1016/0304-3940(87)90073-5 (1987).

14. Koshiyama, H. *et al.* Galanin-induced prolactin release in rats: pharmacological evidence for the involvement of α-adrenergic and opioidergic mechanisms. *Brain research* 507, 321-324, doi:10.1016/0006-8993(90)90290-r (1990).

15. Gera, S. *et al.* First-in-class humanized FSH blocking antibody targets bone and fat. *Proc Natl Acad Sci U S A* 117, 28971-28979, doi:10.1073/pnas.2014588117 (2020).

16. Bekes, M., Langley, D. R. and Crews, C. M. PROTAC targeted protein degraders: the past is prologue. *Nature reviews. Drug discovery*, doi:10.1038/s41573-021-00371-6 (2022).

17. Lu, M., Flanagan, J. U., Langley, R. J., Hay, M. P. and Perry, J. K. Targeting growth hormone function: strategies and therapeutic applications. *Signal transduction and targeted therapy* 4, 3, doi:10.1038/s41392-019-0036-y (2019).

18. Slastnikova, T. A., Ulasov, A. V., Rosenkranz, A. A. and Sobolev, A. S. Targeted Intracellular Delivery of Antibodies: The State of the Art. *Frontiers in pharmacology* 9, 1208, doi:10.3389/fphar.2018.01208 (2018).

19. MacIndoe, J. H. and Turkington, R. W. Stimulation of human prolactin secretion by intravenous infusion of L-tryptophan. *J Clin Invest* 52, 1972-1978, doi:10.1172/JCI107381 (1973).

20. Soto, C. *et al.* Β-sheet breaker peptides inhibit fibrillogenesis in a rat brain model of amyloidosis: implications for Alzheimer's therapy. *Nat Med* 4, 822-826, doi:10.1038/nm0798-822 (1998).

21. Meisl, G. *et al.* Molecular mechanisms of protein aggregation from global fitting of kinetic models. *Nature protocols* 11, 252-272, doi:10.1038/nprot.2016.010 (2016).

22. Rasmussen, C. B. *et al.* Imperfect repeats in the functional amyloid protein FapC reduce the tendency to fragment during fibrillation. *Protein Sci* 28, 633-642, doi:10.1002/pro.3566 (2019).

23. Frankel, R. *et al.* Autocatalytic amplification of Alzheimer-associated Abeta42 peptide aggregation in human cerebrospinal fluid. *Communications biology* 2, 365, doi:10.1038/s42003-019-0612-2 (2019).

24. Andreasen, M. *et al.* Physical Determinants of Amyloid Assembly in Biofilm Formation. *mBio* 10, doi:10.1128/mBio.02279-18 (2019).

25. Kumari, P. *et al.* Structural insights into α-synuclein monomer-fibril interactions. *Proc Natl Acad Sci U S A* 118, doi:10.1073/pnas.2012171118 (2021).

26. Andersen, C. B. *et al.* Branching in amyloid fibril growth. *Biophys J* 96, 1529-1536, doi:10.1016/j.bpj.2008.11.024 (2009).

27. Koloteva-Levine, N. *et al.* Amyloid particles facilitate surface-catalyzed cross-seeding by acting as promiscuous nanoparticles. *bioRxiv*, 2020.2009.2001.278481, doi:10.1101/2020.09.01.278481 (2020).

28. Hartman, K. *et al.* Bacterial curli protein promotes the conversion of PAP248-286 into the amyloid SEVI: cross-seeding of dissimilar amyloid sequences. *PeerJ* 1, e5, doi:10.7717/peerj.5 (2013).